# High-resolution imagery and neural networks link post-tsunami land cover changes to population health and well-being

Eric Peshkin[1], Elizabeth Frankenberg [2] ✉, Peter Katz[1], Cecep Sumantri[3] & Duncan Thomas[1]

As extreme events intensify in force and frequency across the globe, relating the damage and subsequent reconstruction to population health and well-being remains a critical frontier. Here we build a convolutional neural network to classify landcover from satellite images of Indonesia before and after the December 2004 Indian Ocean Tsunami and link those measures to population well-being to demonstrate methods that advance analyses of short-term impacts of extreme events and impacts 5 years later. Population data are from the Study of the Tsunami Aftermath and Recovery (STAR) and 2005 and 2010 censuses. We develop manually labelled training data for eight landcover classes and demonstrate the model performs well using standard metrics. Moreover, measures of change over time in landcover correlate strongly with multiple dimensions of well-being from our household survey data and with aggregate population statistics, both immediately after the event and in the subsequent five years.

Climate change is intensifying the force and frequency of extreme events that damage built and natural environments across the globe, with costs estimated at US$143 billion per year[1]. Beyond damage to property and infrastructure, these events likely have downstream impacts on the health and economic well-being of populations, but scientific evidence documenting these broader impacts is sparse[2,3].

Improving the evidence base requires combining precise measurements of how an extreme event affects the environment with detailed multidimensional assessments of outcomes for exposed populations[4]. Developing tools for measuring both the extent of damage and the pace of recovery and linking these measurements to detailed, accurate data on population outcomes is a critical frontier for understanding how people are affected by environmental exposures and how they recover afterwards[5].

With respect to events' impacts on the environment, satellite images can be used for change detection if imagery is available from before and after the event. In remote contexts these images may be the most reliable source of information. Spatial and temporal coverage of satellites has increased dramatically over the last several decades, but extracting the relevant information and converting it to informative metrics remains complex and time-consuming[6]. Machine learning tools offer new efficiencies for harnessing the power of these data for science[7].

An important gap in this field is linking high-quality, fine-grained measures of destruction and reconstruction extracted from imagery to population-representative data on health and well-being. This paper addresses that gap. We combine high-resolution satellite imagery and machine learning tools with individual-level survey data and community-level census data in the context of a large-scale natural disaster. Our approach demonstrates techniques that enhance scientists' ability to evaluate the immediate and longer-term impacts of extreme events on both the environment and population well-being and to characterize the recovery process. These methods are readily scalable to cover larger geographies and broader populations and relevant for disaster relief efforts both as they unfold and for evaluation once they are completed.

We study the 2004 Indian Ocean Tsunami, which generated devastating flooding in multiple South and Southeast Asian countries[8]. Worldwide casualties totaled over 250,000[9]. The western coastline of the Indonesian province of Aceh was the area hit hardest. Tsunami waves as high as 25–30 m struck Aceh's shore some 15 min after the precipitating earthquake[10], and in Aceh an estimated 160,000 people, roughly 5% of the population, perished[11]. We focus on the tsunami for two reasons. First, it was extremely destructive and unexpected which means that anticipatory behavior does not contaminate interpretation of impacts on population

[1]Duke University, Durham, NC, USA. [2]University of North Carolina at Chapel Hill, Chapel Hill, NC, USA. [3]SurveyMETER, Yogyakarta, Indonesia.
✉e-mail: e.frankenberg@unc.edu

health and well-being. Second, we have collected uniquely rich data about its impacts as part of a population-representative longitudinal survey of individuals and their households who were first interviewed before the tsunami and were followed up annually during the study period. We have maintained the population representativeness of that sample by achieving over 93% re-interview rates of tsunami survivors in the follow-up surveys.

The degree of damage from the tsunami varied spatially. The height, force, and inland reach of water from the tsunami depended on topographical features of the ocean floor and the shoreline[12]. In many areas, low-lying coastal communities were largely destroyed. Along riverways the water encroached as much as 9 kilometers in some areas, versus 3–4 kilometers in other nearby locations[13]. In inland areas, flooding caused substantial damage to the environment but few deaths. Even within small geographic areas the degree of destruction caused by the tsunami varied substantially.

The disaster was followed by a $7 billion reconstruction effort which was, at the time, the largest, most ambitious post-disaster rebuilding effort ever undertaken in a developing country[14]. By many metrics, the "Build Back Better" campaign was successful. Roughly 130,000 homes were built in the five years after the event, to replace the estimated 120,000 that were damaged or destroyed[15]. The United States and Japan supported the reconstruction of a major transportation artery along Aceh's challenging west coast, and many other international, government, and non-government organizations contributed to the rebuilding of infrastructure, including schools, health facilities, village halls, and mosques[14,16].

To measure contextual changes caused by the tsunami and by subsequent reconstruction we implemented a variation of the prominent DeepLabV3+ convolutional neural network (CNN)[17] trained to produce pixel-level segmentations from high-resolution Quickbird satellite imagery that we obtained through a digital imagery grant from MAXAR. We used this approach to construct measures of changes in natural and built structures on the landscape over time. The images include villages and municipalities in the Acehnese districts of Banda Aceh and Aceh Besar, located on the northern tip of the Indonesian island of Sumatra (Fig. 1). The 11 image tiles span a period from six months before the tsunami to 4.2 years afterward. In combination the images cover an area of 839 km$^2$, but the individual images vary in coverage and in size, ranging from 214 km$^2$ to 675 km$^2$. In total, there are over 18 billion pixels in the images.

We created a labelled training set for the CNN, from which the network learned to extract landscape features. After a rigorous evaluation and training program, three team members who lived in the area worked full-time as labelers, in cooperation with our US labelling team. The US and Indonesian team manually labelled a subset of the QuickBird images, demarcating the edges of objects belonging to one of eight categories. The categories reflect the destructive nature of the tsunami (presence of water, building foundations, rubble), underlying geographic and image-specific features (beaches, clouds), and economic assets and activities (buildings, roads and pathways, agriculture). These categories were selected drawing on our on-the-ground knowledge of changes in the built and natural

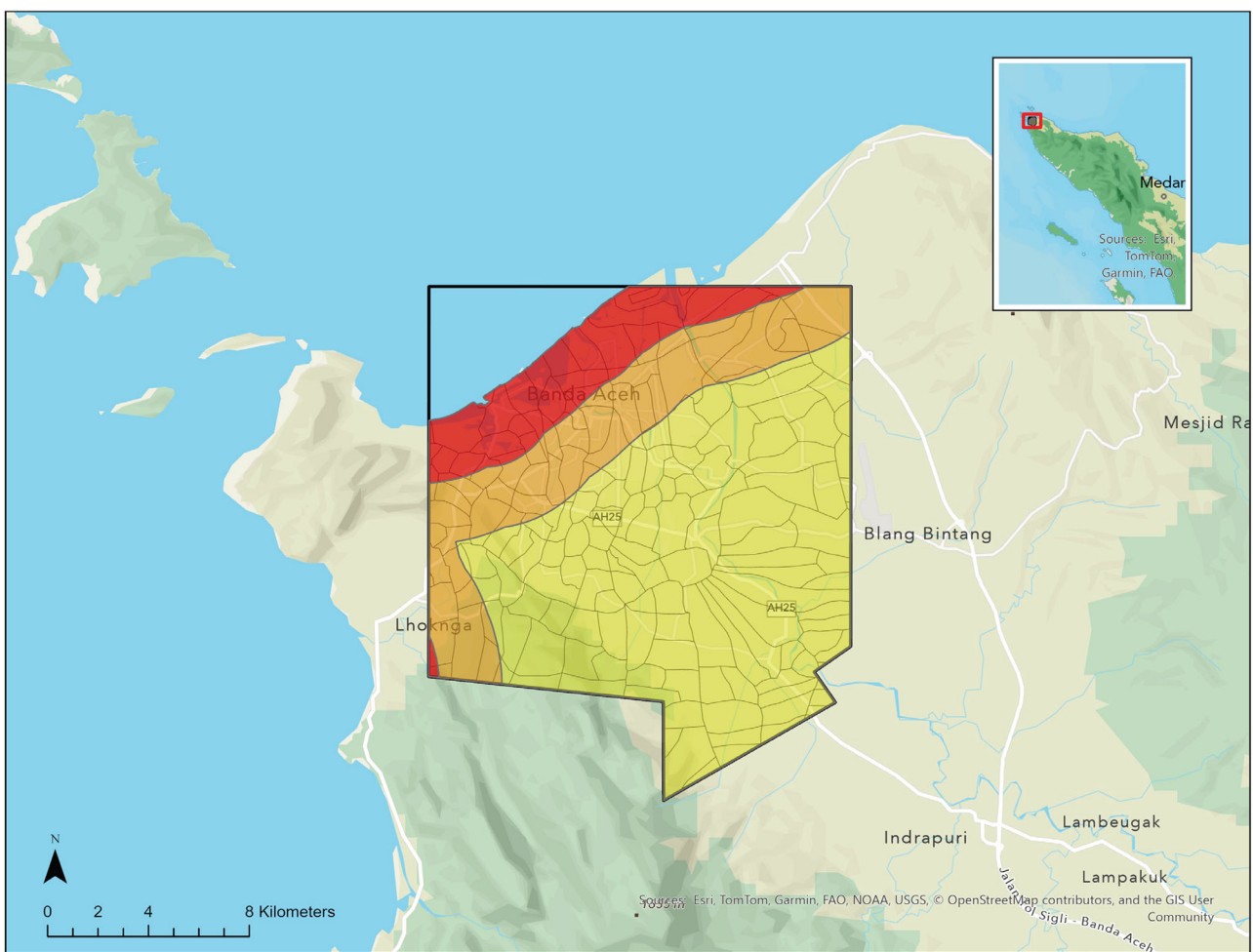

**Fig. 1 | Map of study area with zones demarcated based on proximity to the coast.** Map of the area of Aceh Province, Indonesia, for which high resolution imagery is available for four time points (June 2004, late December 2004, July 2007, February 2009). The area is 289 km$^2$ and contains 164 administrative areas (villages or municipalities, outlined in black). Zones are designated by proximity to the shore (red is within 4.3 km, orange is 4.3–8.3 km, yellow is 8.3 km or more). The map inset shows the area in relation to the northern end of the island of Sumatra. Sources: Esri, Tom Tom, Garmin, FAQ, NOAA, USGS, © OpenStreetmap contributors, and the GIS User Community.

**Fig. 2 | MAXAR imagery of a neighborhood in Banda Aceh and Output from the CNN Segmentation. A** Quickbird imagery for a neighborhood in Banda Aceh, Indonesia, before and at multiple time points after the tsunami. **B** segmentations produced by our CNN. Classifications are other (grey), agriculture (dark green), water (dark blue), rubble (brown), foundation (orange), beach (yellow), cloud (white), road (light blue) and building (magenta). Destruction is visible in the images from December 2004 and August 2005. Over time, structures are rebuilt. Satellite images courtesy of the DigitalGlobe Foundation (now MAXAR).

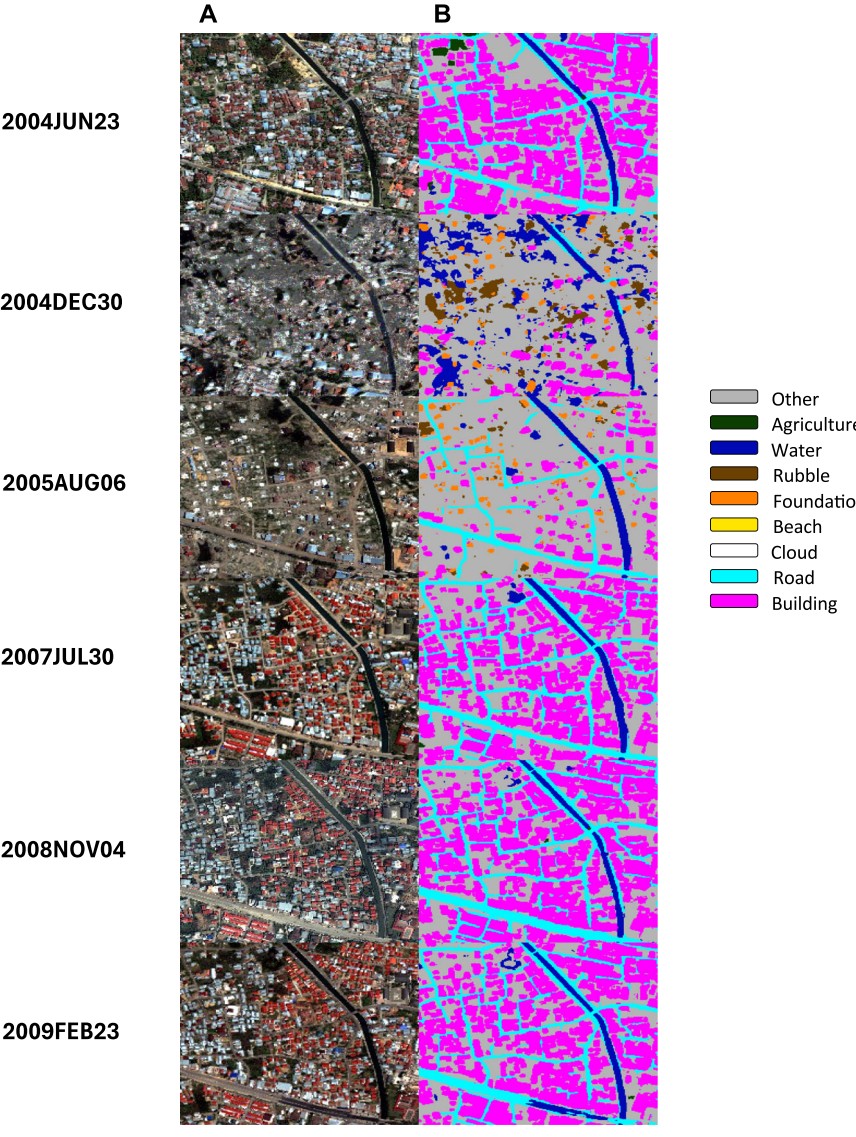

environment because of the tsunami and during reconstruction. All labels were cross-validated to ensure accuracy and consistency. Pixels not contained within a labelled object were assigned to a catch-all class "other" which includes, for example, unimproved land. Once trained, the CNN classifies and localizes objects from each of the eight categories at a per-pixel level across multiple time points. We use the pixel-specific assignments to construct measures of landcover at each time point, which we use to analyze patterns of change in landcover over time. The data provide high-resolution measures of both the degree of destruction caused by the tsunami and the extent of subsequent rebuilding. Figures 2 and 3 display examples of the input images and the resulting segmentations, demonstrating the method at a small neighborhood scale (Fig. 2) and a much larger regional scale (Fig. 3). In both the input images and the segmentations, change near the coast from before to just after the tsunami is apparent. Water, rubble, and foundations are more prevalent post-tsunami, while buildings are less prevalent. Rebuilding over time in the period after the disaster is also clear as buildings are reconstructed (in Fig. 3), and growth also extends into areas that were primarily agricultural pre-tsunami.

We link our measures of land cover to two primary sources of population data that have complementary strengths. One population data source is census data, collected by Statistics Indonesia in 2005 as part of the Aceh-Nias Special Census and then again in 2010 as part of the regular decennial census[18,19]. The other source of population data is a survey that we designed

and collected, the Study of the Tsunami Aftermath and Recovery (STAR), which is a longitudinal survey of individuals, households, and communities interviewed before and at multiple points after the tsunami[20]. STAR builds off the 2004 National Socioeconomic Survey (SUSENAS) conducted by Statistics Indonesia 10 months before the tsunami that provides a pre-tsunami baseline. The censuses and the STAR survey provide data on individuals living throughout the province of Aceh. In this paper we focus on the population from the subset of communities in the districts of Banda Aceh and Aceh Besar for which the high-resolution imagery is available at four time points before and after the December 2004 tsunami, between June 2004 and February 2009.

The census data are cross-sectional and involve relatively short questionnaires but are available for a larger number ($N = 164$) of villages and municipalities. STAR, a longitudinal survey, followed up all participants in the pre-tsunami baseline, ascertaining survival status for over 98%, tracking displaced survivors, and interviewing them (and their children born after the tsunami) annually for 5 years after the disaster and again 10 and 15 years after the disaster. Our analyses draw on STAR data from the first (2005) and fifth (2009) follow-up surveys after the tsunami.

We analyze data for 2947 STAR respondents who, in 2004, were living in the 43 communities for which high-resolution imagery is available over time. Attrition poses special challenges in a study of a major disruptive event. It is critical to measure the impact of the disaster on a sample that is

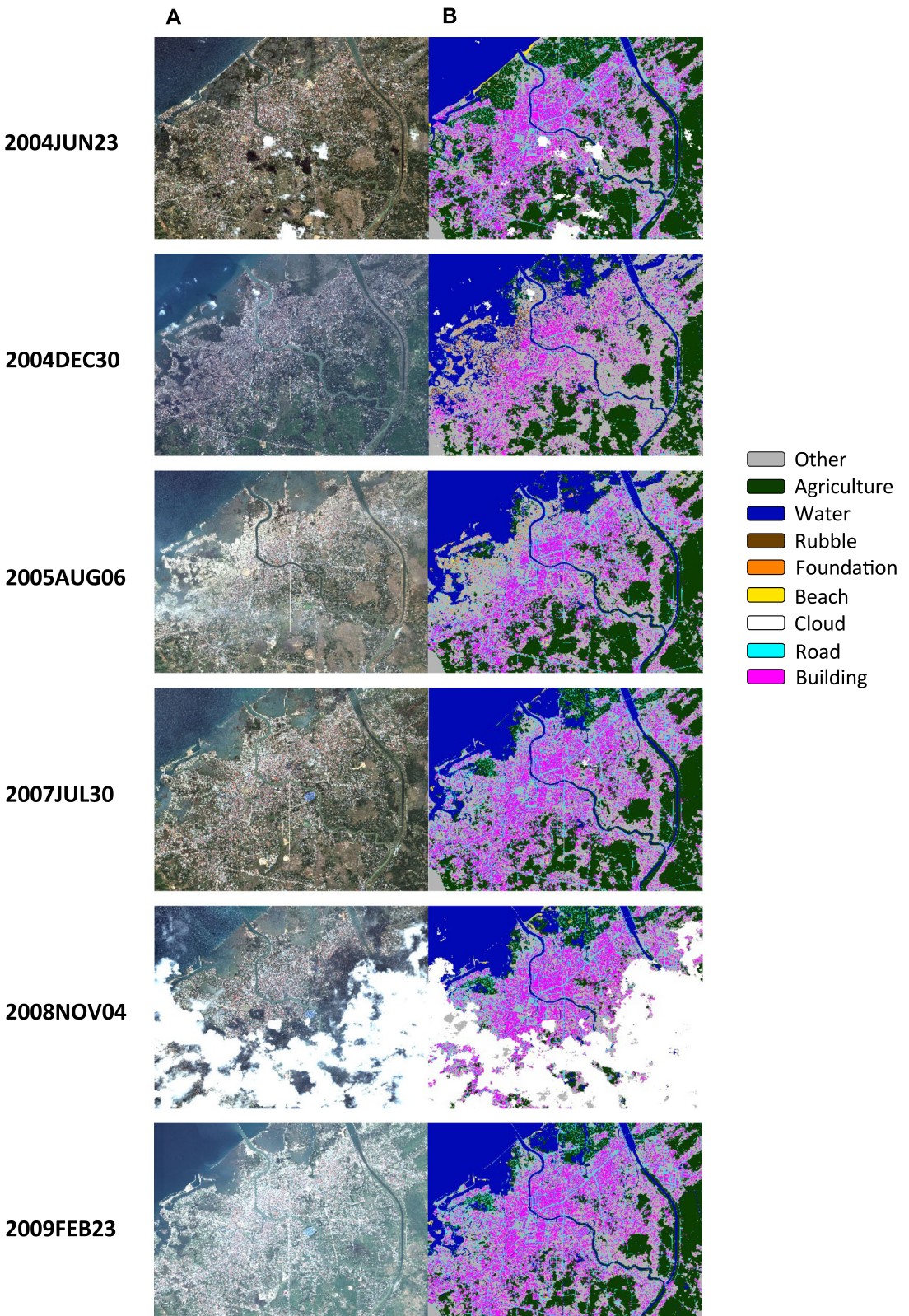

**Fig. 3 | MAXAR imagery of Banda Aceh and Output from the CNN Segmentation.** **A** Quickbird imagery for the city of Banda Aceh, Indonesia, before and at multiple time points after the tsunami. **B** displays the segmentations produced by our CNN. Classifications are other (grey), agriculture (dark green), water (dark blue), rubble (brown), foundation (orange), beach (yellow), cloud (white), road (light blue), and building (magenta). Destruction is visible in the images from December 2004 and August 2005. Clouds obscure much of the image from 2008. Over time, structures are rebuilt and increase as a share of landcover. Satellite images courtesy of the Digital Globe Foundation (now MAXAR).

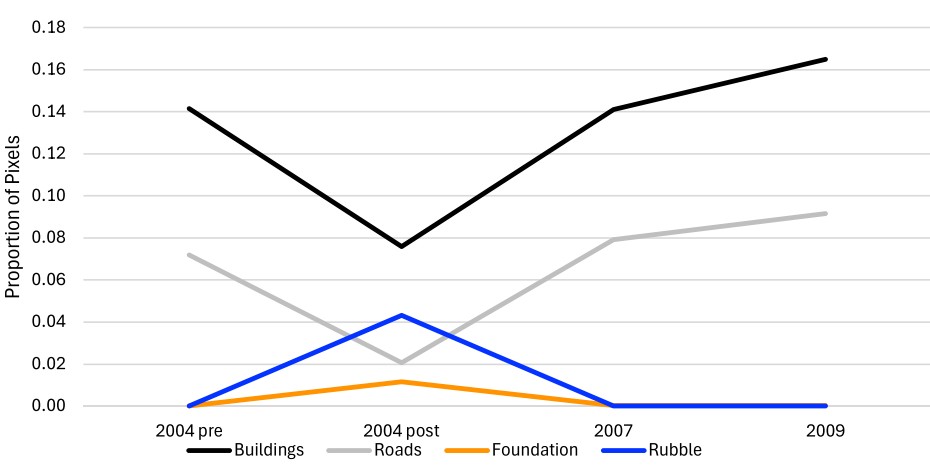

**Fig. 4 | Changes over time in Markers of the Built Environment in the Coastal Zone (< 4.3 km from shore).** Changes over time in the proportions of pixels in the coastal (< 4.3 km from the shore) zone classified as buildings (black), roads (grey), foundation (orange), or rubble (blue). Proportions are based on classification of over 150 million pixels at each time point, generating extremely small standard errors (< 0.00005). Share, standard errors, and numbers of pixels are reported in Panel A of Table 1.

representative of the entire post-tsunami population, including those who moved. By designing, testing, and implementing sophisticated follow-up procedures, we have successfully re-interviewed over 96% of the survivors and our study sample, regardless of their post-disaster locations. The survey, therefore, is representative of the population of survivors immediately after the tsunami. In addition, the STAR questionnaires cover multiple inter-related topics, including physical and psychosocial health and economic outcomes for all surviving members of the original households.

This direct linkage of changes in the environment to changes in population health and well-being is a key frontier for advancing the value to science and policy of these methods and translating advances in measurement to improvements in the lives of those affected by disasters. Studies have pointed out the potential of satellite imagery for assessing the recovery process[21–24]. Approaches in these studies include short-term assessments of the recovery status of damaged buildings, mapping of the return of agriculture and settlements, and inference that certain patterns of change in land cover and land use imply population recovery. However, in the absence of direct empirical tests of the relationships between the imagery and measures of population well-being, the existence and importance of the hypothesized relationships have not been established. Our results provide compelling evidence that image-based measures of land cover changes, both immediately after a disaster and over the next several years, are strongly associated with markers of health and well-being for individuals and with demographic change at the community level.

## Results
### Changes in land cover
We use the QuickBird imagery to describe changes in landcover over time, focusing on results from four points in time: six months before the tsunami (June 23, 2004), 2–4 days after the tsunami (December 28 or 30, 2004), two and half years after the tsunami (July 30, 2007) and slightly over 4 years post-tsunami (February 9 or 23, 2009). Results from our accuracy assessment are reported in the Methods section.

As an initial summary of changes, we create three geographic zones based on distance from the coast (Fig. 1). Each zone contains 54 or 55 of the 164 administrative areas included in the imagery (administrative area boundaries are outlined on the map and are classified based on the distance of their geographic center from the coast). The inland zone is made up of administrative areas whose center is at least 8.3 km from the coast, whereas for the coastal zone, the centers are less than 4.3 km from the shoreline. Those in the range of 4.3 to 8.29 km to the shore fall into the mid zone.

For each of the 164 administrative areas we process the segmented output from our CNN to compute land cover shares within the area demarcated by the boundaries. We construct statistics for each zone by aggregating the classifications across all administrative units in the zone. For each zone and time point our classifications are based on between 284 and

417 million pixels. Because the estimates of land cover are based on large numbers of pixels, the standard errors are very small (< 0.00005). Among our eight classes (water, foundations, rubble, buildings, roads, agriculture, beaches, and clouds), some classes are strongly positively or negatively correlated over time. We illustrate this pattern for the coastal zone in Fig. 4, which also conveys the dramatic land-cover changes in this zone from the tsunami: changes in buildings and roads are positively correlated, and both are negatively correlated with changes in rubble (which is weakly positively correled with foundations). The shares of pixels classified as buildings or roads fall dramatically just after the tsunami, then rise again during reconstruction, eventually surpassing their pre-tsunami levels. The reverse happens for rubble and foundations.

Given the strengths of the correlations between classes, we focus on three classes that together and in the context of the tsunami, reflect important dimensions of the built and natural environment: water, agriculture, and buildings. These classes account for between 48% and 64% of land cover across the three zones. Figure 5 presents temporal and spatial variation in land cover for these three classes, standardized to sum to 100% (summary statistics and standard errors are presented in Table 1). For the inland zone (panel A) that was not damaged by the tsunami, the distribution of pixels changes very little across the four time points. Agriculture pre-dominates at each time point, accounting for over 90% of the pixels classified into these three groups. For the middle distance zone (panel B), land cover classified as water triples just after the tsunami, but water coverage returns to roughly its pre-tsunami level by 2007. After the tsunami, buildings account for increasingly larger shares as time goes by, while agriculture accounts for smaller shares.

The drama of tsunami destruction and reconstruction is evident in the coastal zone (panel C). Water coverage increases dramatically just after the tsunami, from 8% to 58% of the pixels in these three classes, while buildings and agriculture decrease. By July 2007, water coverage is much lower (though markedly higher than before the tsunami, consistent with erosion of the original shoreline). Shares of buildings and agriculture, meanwhile, both rise— buildings to a slightly higher share than before the tsunami. Over the next 20 months buildings increase while agriculture decreases slightly. These graphs quantify the changes that are visible in the imagery and segmentations in Figs. 2 and 3 and provide evidence that the patterns of change captured by our segmentations differ across zones delineated by distance from the shore in ways that are consistent with the nature of the disaster.

### Social and Economic Outcomes at the Community and Individual Levels
We also analyze how the land cover measures derived from high resolution satellite imagery vary with demographic, health, and economic outcomes at the community and individual levels. We examine outcomes in 2005, which reflect the short-term impacts of the tsunami, and changes in outcomes

**Fig. 5 | Distribution of Pixels Over Time by Distance from Coast.** Water, Buildings, and Agriculture Distribution of landcover across water (blue), buildings (red) and agriculture (grey), over time, by distance from the coast. **A** In the inland zone little change occurs over time. **B** In the mid-zone buildings increase in prominence over time. **C** In the coastal zone water rises then falls, whereas the reverse occurs for buildings and agriculture. Shares are constructed by aggregating across all pixels classified for communities that lie in each zone. Shares, standard errors, and numbers of pixels are reported in Panel B of Table 1.

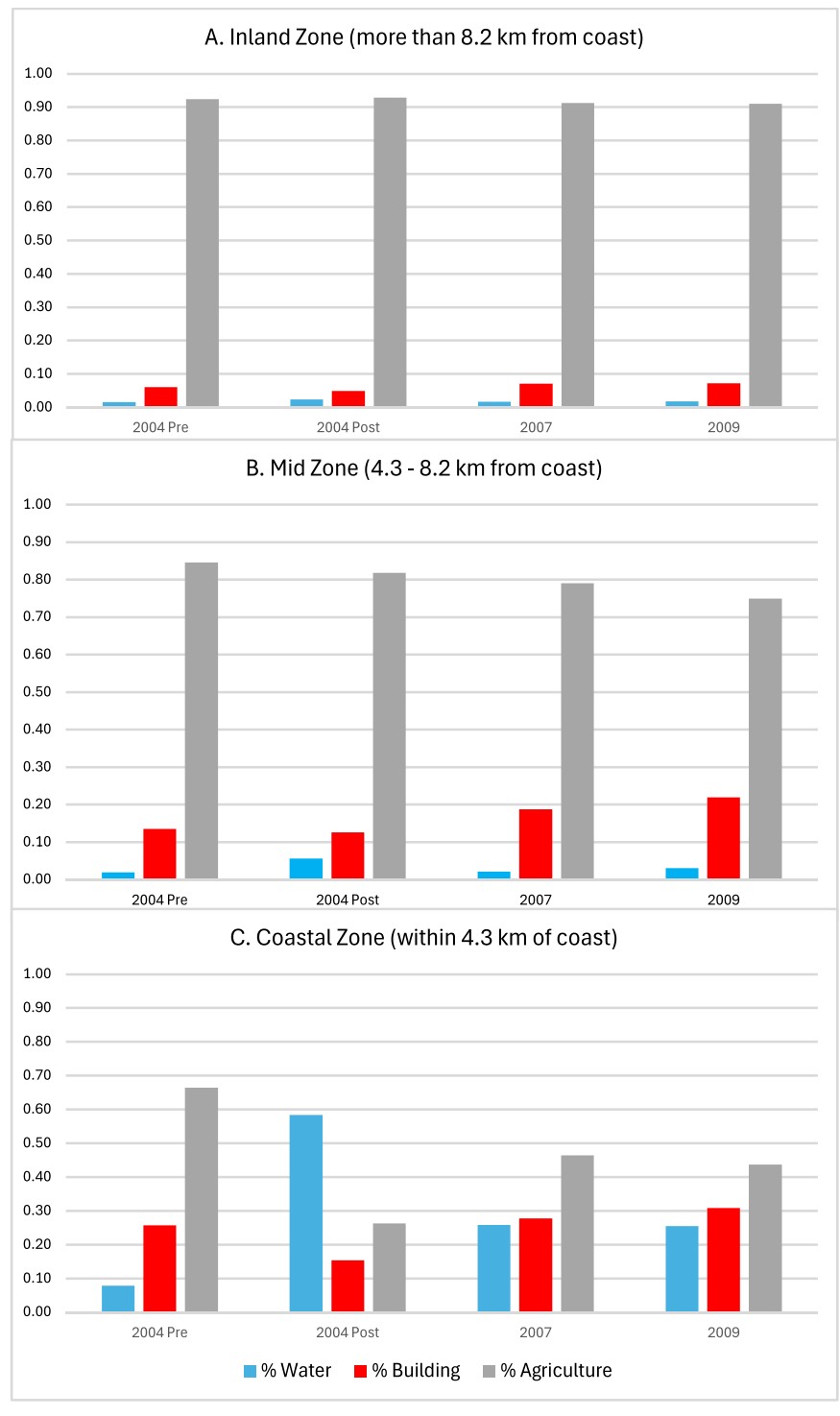

between 2005 and 2009–10, during which recovery assistance flowed into the study area. The 2005 Special Census of Aceh and Nias provides data on population size for each of the 164 communities enumerated shortly after the tsunami.

Using STAR data we construct a measure of mortality during the tsunami for 2947 individuals who lived in the study area in 2004 and participated in the pre-tsunami baseline survey. For 2065 survivors who were interviewed in the first follow up survey (STAR1) in 2005 we examine displacement as a result of the tsunami. Lastly, for 1109 respondents who were 15 or older at STAR1 and provided face-to-face interviews as part of both the STAR1 (2005) and the STAR5 (2009) surveys, we analyze self-

reported levels of post-traumatic stress reactivity and change in SES as measured by their self-assessed step on a 6-step ladder (from poorest at the bottom to richest at the top) in the year after the tsunami, relative to just before the tsunami. Table 2 presents summary statistics.

We use linear regression to model short-term outcomes as a function of the proportions of land cover accounted for by water, buildings, and agriculture, measured six months before the tsunami and again 2–4 days after the tsunami. In these models shares are computed based on pixel classifications from either the entire administrative area (for the census data) or (for the STAR data) from within a buffer with a 500 m radius positioned around the center of the survey enumeration area. We interpret the

## Table 1 | Landcover Shares by Zone and Time Period

| | | 2004 Pre | 2004 Post | 2007 | 2009 |
|---|---|---|---|---|---|
| A. | | Markers of the built environment in the coastal zone (see also Fig. 4) | | | |
| | Buildings | 0.1416 | 0.0758 | 0.1412 | 0.1650 |
| | *Standard Error* | *0.000020* | *0.000015* | *0.000019* | *0.000021* |
| | Roads | 0.0719 | 0.0206 | 0.0791 | 0.0915 |
| | *Standard Error* | *0.000015* | *0.000008* | *0.000015* | *0.000016* |
| | Foundation | 0.0001 | 0.0116 | 0.0004 | 0.0003 |
| | *Standard Error* | *0.000001* | *0.000006* | *0.000001* | *0.000001* |
| | Rubble | 0.0001 | 0.0432 | 0.0001 | 0.0002 |
| | *Standard Error* | *0.0000005* | *0.000011* | *0.000001* | *0.000001* |
| | N | 316,269,120 | 331,707,488 | 342,511,232 | 317,413,952 |
| B. | | Water, buildings, and agriculture over time, by distance from the coast (see also Fig. 5) | | | |
| >8.2 km from coast | | | | | |
| | Water | 0.015 | 0.023 | 0.017 | 0.017 |
| | *Standard Error* | *0.000004* | *0.000005* | *0.000005* | *0.000005* |
| | Buildings | 0.061 | 0.048 | 0.071 | 0.072 |
| | *Standard Error* | *0.000012* | *0.000012* | *0.000014* | *0.000014* |
| | Agriculture | 0.924 | 0.929 | 0.913 | 0.910 |
| | *Standard Error* | *0.000034* | *0.000039* | *0.000037* | *0.000036* |
| | N | 210,359,520 | 158,128,496 | 184,926,016 | 191,509,264 |
| 4.3–8.2 km from coast | | | | | |
| | Water | 0.019 | 0.056 | 0.022 | 0.031 |
| | *Standard Error* | *0.000006* | *0.000009* | *0.000006* | *0.000007* |
| | Buildings | 0.135 | 0.126 | 0.187 | 0.220 |
| | *Standard Error* | *0.000020* | *0.000020* | *0.000024* | *0.000026* |
| | Agriculture | 0.846 | 0.818 | 0.791 | 0.750 |
| | *Standard Error* | *0.000040* | *0.000041* | *0.000041* | *0.000041* |
| | N | 151,303,840 | 144,814,720 | 143,099,376 | 136,006,944 |
| <4.3 km from coast | | | | | |
| | Water | 0.079 | 0.583 | 0.258 | 0.255 |
| | *Standard Error* | *0.000011* | *0.000025* | *0.000018* | *0.000019* |
| | Buildings | 0.257 | 0.154 | 0.278 | 0.308 |
| | *Standard Error* | *0.000026* | *0.000021* | *0.000026* | *0.000028* |
| | Agriculture | 0.664 | 0.262 | 0.464 | 0.437 |
| | *Standard Error* | *0.000036* | *0.000026* | *0.000032* | *0.000032* |
| | N | 174,307,088 | 163,131,776 | 174,123,200 | 170,041,456 |

coefficients corresponding to the post-tsunami shares as indicating the association between the level of an outcome in 2005 and an increase in the share of a particular class of land cover just after the tsunami relative to before the tsunami. Table 3 reports regression results.

At the 2005 post-tsunami census, the mean population size in the communities we analyze was 1,491 (Table 2). Examining the log of population size in 2005 (Table 3), as the relative share of water increases just after the tsunami, population size decreases—a consequence of both the tsunami's mortality toll and the displacement of survivors. (The coefficients on post-tsunami share of land devoted to buildings and agriculture are positive and for buildings statistically significant at $p = 0.08$).

We turn next to results for individual outcomes, drawing on the STAR data. These models include controls for age, sex, educational level, and district of residence, to distinguish the urban district of Banda Aceh from the rural district of Aceh Besar. Roughly one in five of the individuals interviewed in 2004 was killed in the tsunami (Table 2). Immediate post-tsunami increases in water are strongly and positively associated with the probability of death, whereas a relatively greater share of buildings post-tsunami is

associated with a decrease in the probability of death (Table 3). Both coefficients are statistically significant.

About 80% of STAR respondents survived the tsunami. These survivors faced a decision regarding whether to remain at the location of their pre-tsunami residence or go elsewhere. Just under half (45.6%) of the respondents reported being displaced after the disaster. Increase in water share is strongly and positively associated with the risk of displacement (Table 3).

Among survivors the psychological costs of experiencing the tsunami were high[25]. The average value of our measure of post-traumatic stress reactivity is 7.7, out of a maximum of 21 (Table 2). Land cover shares are strongly related to the intensity of symptoms of post-traumatic stress (Table 3). The coefficient for water post-tsunami is large and positive, although it is imprecisely estimated. The coefficients for building and agriculture shares are large and negative, indicating better psychological health in the year after the tsunami among those whose communities experienced less damage. Both coefficients are statistically significant.

**Table 2 | Summary Statistics for Population Data**

| | | Mean | Std Error* | Range | Std Dev |
|---|---|---|---|---|---|
| Community | Population size in 2005 (N = 164 communities) | 1,491 | | 49–7046 | 1523 |
| Individual | % Dead (N = 2947 baseline respondents) | 19.8 | 0.046 | | |
| Individual | % displaced (N = 2065 survivors in 530 ivwed hh) | 45.6 | 0.05 | | |
| Individual | Level of post-traumatic stress reactivity STAR1 (2005) | 7.7 | 0.28 | | |
| Individual | Change in position on a 10 step SES ladder, pre versus just post tsunami | −0.18 | 0.03 | | |
| Individual | Change in post-traumatic stress reactivity, STAR5 versus STAR1 | −5.5 | 0.034 | | |
| Individual | Change in ladder position, 2009 relative to 2005 | 0.52 | 0.038 | | |
| Community | Population growth rate, 2005–2010 (N = 164 communities) | 0.09 | | -0.11-0.61 | 0.155 |
| Individual | % Male | 50 | 0.007 | | |
| Individual | Years of education (if 25 years or older) | 10.7 | 0.28 | | |
| Individual | Age (years) | 28.4 | 0.44 | | |

* standard errors adjusted to account for clustering

Lastly, on average, perceptions of SES declined after the tsunami, with respondents reporting themselves on a lower SES step in the year after the tsunami than in the year before the tsunami. Self-perceived SES falls among individuals from communities where the share of water rose just after the tsunami (the coefficient on water is statistically significant) but rises for individuals from communities where the share of buildings rose.

For all outcomes measured in the months after the tsunami, the relationships with land cover just after the tsunami, holding constant shares six months before the tsunami, are strong. The post-tsunami land cover shares are jointly significant in each model, as indicated by the F statistics. Additionally, the $R^2$ statistics, which indicate the share of variation in outcomes explained by the model overall, are high, ranging from 0.10 for post-traumatic stress to 0.47 for mortality. In these models water share is most consistently related to the short-term outcomes but shares of buildings and agriculture also matter. The results indicate the importance of accounting for multiple components of land cover change rather than focusing solely on one component, such as building destruction.

Documenting how outcomes for the population are affected by the disaster's immediate impact is important, but communities and individuals continue to experience change in the years after an extreme event. In the aftermath of the tsunami countries throughout the region undertook recovery efforts, funded in part by donations from around the globe[14].

Figures 2–5 provide evidence of the impact of rebuilding, particularly in the areas close to the shoreline, where destruction was most intense. Post-tsunami, in the coastal zone the shares of land devoted to buildings and agriculture rose relative to the share accounted for by water. We now examine whether post-tsunami changes (between 2005 and 2009) in individual and community outcomes are significantly related to post-tsunami changes in land cover over that period.

We begin with the change in post-traumatic stress reactivity, constructed by subtracting the 2005 PTSR score from the 2009 score. On average the score declined 5.5 points, indicating improvement in psychosocial health (Table 2). The signs on the coefficients associated with changes in shares of buildings and agriculture are negative, and both coefficients are statistically significant (Table 3). Individuals from communities where buildings and agriculture increase in prevalence over time report relatively larger reductions in post-traumatic stress reactivity between the 2005 and 2009 interviews.

For the SES ladder, on average respondents report being on a higher step in 2009 than in 2005. The mean change is 0.52 (Table 2). The negative coefficient on water shares indicates that relative decreases in water shares between 2005 and 2009 are associated with more positive changes in perceptions of SES, (this relationship is statistically significant). Changes in shares devoted to buildings and agriculture are unrelated to changes in perceived SES.

We also analyze correlations between changes in land cover and the community-level population growth rates, constructed from community-level data from the 2005 Special Census and the 2010 decennial census. The coefficient on the water share is negative: increases in water shares are associated with lower rates of population growth, whereas decreases are associated with higher growth rates. The opposite is true for changes in buildings: increases in building shares are associated with higher rates of population growth. Both coefficients are statistically significant.

Our results confirm the strength of the relationships between changes in landcover, brought about first by the tsunami and subsequently by the recovery program, and a series of indicators of well-being measured at the individual and community levels. In all models the variables measuring change in land cover over the 2005 to 2009 period are jointly significant as shown by the F test statistics.

We compare these results to results derived from more standard approaches to measurement of landscape variation from imagery, such as nightlight data and the Normalized Difference Vegetation Index (NDVI) (Supplementary Table 1). NDVI, which measures vegetation greenness, has been used to detect damage to vegetation from flooding and wildfires[26,27]. Drawing on NASA's MODIS products, we estimated models that parallel those reported in Table 3 but that use community-specific NDVI values in 2004, 2005, and 2009[28]. The results are substantively similar across the individual-level outcomes we consider, but explain considerably less of the variation in outcomes than the models we estimated using segmentations produced by our CNN. For example, with respect to mortality, the $R^2$s are 0.47 (CNN) versus 0.31 (NDVI).

Other work has also analyzed changes in intensity of nightlights as indicative of damage and recovery in the aftermath of extreme events[29]. An analysis that combined data on nightlight brightness from the Defense Meteorological Satellite Program Operational Linescan System with data from the wider set of districts in which STAR was conducted found that there were significant relationships between nighttime imagery brightness and per capita spending levels aggregated to the community level[30]. Here we focus on individual-level outcomes for a smaller and more geographically concentrated set of communities. The results from estimating our models with nighttime light intensity as the marker for damage and recovery (Supplementary Table 1) are difficult to interpret, most likely because they lack the spatial resolution required to precisely chart localized destruction and reconstruction across communities that are close to one another[31].

**Table 3 | Regression Results, Outcomes in 2005 and Changes in Outcomes (2009-2005)**

### 3 A. Outcomes in 2005

| | Census | Individual Outcomes* | | Post Traumatic | SES ladder |
| --- | --- | --- | --- | --- | --- |
| | Pop Size (ln) 2005 | Killed in tsu. | Displaced | Stress Reactivity | Post-Pre |
| pct Water post tsunami | −2.22 | 1.16 | 1.80 | 2.63 | −1.02 |
| | (1.02) | (0.48) | (0.28) | (2.78) | (0.30) |
| | [0.030] | [0.020] | [0.000] | [0.350] | [0.002] |
| pct Buildings post tsunami | 2.30 | −0.85 | −0.11 | −5.30 | 0.75 |
| | (1.32) | (0.24) | (0.29) | (1.77) | (0.25) |
| | [0.081] | [0.001] | [0.716] | [0.005] | 0.004 |
| pct Agriculture post tsunami | 0.08 | −0.06 | 0.11 | −4.59 | 0.02 |
| | (0.65) | (0.15) | (0.17) | (1.25) | (0.12) |
| | [0.904] | [0.701] | [0.499] | [0.001] | [0.844] |
| pct Water pre tsunami | −0.50 | −0.73 | −1.66 | −8.37 | −0.14 |
| | (2.62) | (1.01) | (0.70) | (7.54) | (0.69) |
| | [0.847] | [0.475] | [0.023] | [0.274] | [0.845] |
| pct Buildings pre tsunami | −3.21 | 0.58 | 0.81 | −1.49 | −0.52 |
| | (1.18) | (0.34) | (0.39) | (2.26) | (0.37) |
| | [0.006] | [0.095] | [0.042] | [0.512] | [0.173] |
| pct Agriculture pre tsunami | 0.21 | −0.32 | −0.56 | −5.50 | 0.61 |
| | (0.56) | (0.28) | (0.38) | (2.13) | (0.34) |
| | [0.703] | [0.267] | [0.147] | [0.013] | [0.078] |
| Constant | 7.33 | 0.19 | 0.24 | 10.34 | -0.28 |
| | (0.51) | (0.08) | (0.19) | (0.98) | (0.18) |
| Observations | 164 | 2,947 | 2,065 | 1,109 | 1,109 |
| $R^2$ | 0.280 | 0.468 | 0.232 | 0.098 | 0.113 |
| $X^2$/F stat, LC post-tsu | 12.75 | 41.79 | 26.82 | 9.09 | 23.44 |
| $P$-value($X^2$ / F) | 0.005 | < 0.000 | < 0.000 | < 0.000 | < 0.000 |

### 3B. Change in Outcomes over Time

| 2009-05 | Individual Outcomes* | | Census: |
| --- | --- | --- | --- |
| | Post Traumatic Stress Reactivity | SES ladder | Population Growth Rate |
| Change in Water | −1.28 | −0.79 | −0.394 |
| | (2.49) | (0.35) | (0.201) |
| | [0.611] | [0.028] | [0.050] |
| Change in Buildings | −6.75 | 0.62 | 0.475 |
| | (2.35) | (0.49) | (0.183) |
| | [0.006] | [0.215] | [0.009] |
| Change in Agriculture | −4.53 | −0.12 | −0.164 |
| | (1.29) | (0.14) | (0.074) |
| | [0.001] | [0.389] | [0.026] |
| Constant | −4.06 | 0.45 | 0.072 |
| | (0.68) | (0.19) | (0.041) |
| Observations | 1,109 | 1,109 | 164 |
| $R^2$ | 0.035 | 0.018 | 0.210 |
| $X^2$/F stat, LCC 2009-2005 | 4.79 | 2.56 | 7.00 |
| p($X^2$ / F) | 0.001 | 0.025 | 0.030 |

Coefficients, clustered standard errors (in parentheses), *p*-values [in square brackets]
*controls for sex, age, education, and district

## Discussion

These results provide compelling evidence that land cover changes, both immediately and in the several years after a disaster, are informative indicators of destruction and reconstruction that are strongly associated with a range of markers of well-being for individuals and with demographic change at the community level. The measures of land cover change are constructed by implementing a principled image processing and CNN pipeline that produces semantic segmentations from high resolution satellite imagery, and that is readily scalable.

This method efficiently extracts information from satellite imagery that we use to construct local, area-specific measures of the 2004 Indian Ocean tsunami's immediate impact on landcover and measures of the evolution of landcover as funds for reconstruction and rebuilding flowed into the region. These advances in measurement expand the scientific toolkit for characterizing contextual change resulting from many different forces, including climate change and war.

Our work extends the literature on the application of CNNs to land cover change in the context of disasters in important ways. We consider 8 classes of landcover in urban, rural, and coastal neighborhoods simultaneously, over many timesteps, and during a period where a major disaster and subsequent recovery efforts created enormous heterogeneity across images. By considering multiple classes that in combination, reflect disaster impacts and economic activity, we more completely characterize the complexity of destruction and reconstruction than is typical. Second, we capture meaningful variation in change across communities that are close to each other, which is important in the context of disasters because impacts often differ within localized areas and are difficult to predict with precision beforehand.

A key innovation is that we combine the information we extract from satellite imagery with surveys and census data to describe how populations are affected by and recover from a major disaster—an approach that is currently underexploited in the context of studies of economic development[32]. The linkage of comprehensive, high-quality data from censuses and a longitudinal survey with landcover measures constructed from segmentations produced using machine learning techniques is rare. Our results demonstrate the value-added of our contextual measures for both the immediate outcomes of survival and displacement and for psychosocial health and socioeconomic status in the year after the event. We also analyze changes in outcomes over the five years after the disaster, as reconstruction funds were deployed to "build back better." We show that changes in land cover, measured post-disaster, are relevant for the recovery process of individuals and communities, just as destruction from the disaster matters in its immediate aftermath. Moreover, with additional imagery, our investments in a labelled data set and model development can be harnessed to provide insights into many other short- and longer-term behaviors in the aftermath of a disaster, including characterizing sequential destination sites for migrants and the displaced, as well as the time path of return migration.

Most work that applies machine learning to the analysis of natural disasters focuses on identifying places that experienced destruction from a particular event and assessing the extent of that destruction. This application is important and can inform rescue and humanitarian aid strategies, particularly as datasets of labels become more available, rapid access to high-resolution imagery expands, and analysts gain proficiency and speed at combining these two inputs. Potentially, these methods could increase the accuracy with which areas with the relatively greatest physical damage are identified, or detect areas where people have erected temporary shelter, both of which could improve accuracy in targeting assistance.

A contribution of this study is that it provides a real-world illustration of the value of linking measures of damage to the built and natural environment to high-quality measures of the experiences of exposed populations both immediately after the disaster and over the longer-term. This is important because immediate impacts differ from longer-term impacts, and those differences provide insights into the population groups that are most vulnerable in the aftermath of a disaster. Although some work addresses recovery by characterizing trajectories of changes in landcover over time for landscapes affected by disasters[22,26,33], almost no other work directly links measures of variation in land cover to measures of variation in population well-being or considers how these relationships evolve over multiple years as reconstruction efforts take place. Expanding attention to population well-being and later phases of the post-disaster period is fundamental if the methods are to realize their full potential.

These insights have important implications for science and policy. For example, in the aftermath of a disaster, information is at a premium, and these methods can help identify communities in greatest need, including those entirely cut off, as well as routes for evacuation and for provision of assistance. Estimation of the size of the affected population can be improved with these methods, which in turn will provide a better foundation to assess immediate needs and estimate the costs of damage areas affected by the disaster. It will be possible to inform real-time evaluations of assistance as it is delivered immediately after disasters and guide dynamic allocation of assistance, ensuring that investments are as effective as possible in supporting the affected populations. Over the longer-run, with these methods, it is possible to document the actual timeline of rebuilding of the built environment, including roads, bridges, and housing along with the re-emergence of agricultural cultivation and other sources of livelihoods and provide a rigorous evaluation of the assistance programs. More broadly, merging information on changes in land cover and land use with data on population measures of socioeconomic status and well-being provides a far richer and more comprehensive picture of immediate impacts of a major disaster and the longer-term trajectories of well-being than does either source alone.

The methods we use are not without limitations. Implementation of a study that measures impacts of a disaster over the long term raises several challenges. These include the need for cloud-free imagery measured before and at key time points after the disaster, well-curated labelled training data, and the collection of high-quality population-representative data on individual well-being immediately after the disaster and over the long-term. A limitation of this study is that we consider one type of event within a relatively small area. The generalizability of the evidence to a broader set of events and other areas is not known. Evaluating the approach in the context of other events and in other settings will be an important contribution to science. In addition, new methods that improve efficiency are continually being developed and may be relevant for the questions we address[34,35].

## Methods

We train and implement a modified version of the DeepLabV3 +[17] CNN to produce semantic segmentations of satellite imagery that classify each pixel into one of eight landcover classes: rubble, foundation, water, beaches, clouds, buildings, roads, and agriculture. After examining the correlations among this relatively comprehensive set of eight classes, we selected three: water, buildings, and agriculture, to relate to population outcomes constructed from census and survey data.

### Imagery and Training Dataset

We process 11 georeferenced QuickBird images. These high-resolution images span the period from June 23, 2004 to February 23, 2009 and capture the northern end of the island of Sumatra, focusing on the city of Banda Aceh and parts of the surrounding more rural district of Aceh Besar. Because the images vary with respect to the geographic area covered, the number of times a particular location appears varies as well. Each image was initially provided by DigitalGlobe (now MAXAR) in the form of one multispectral and one panchromatic image spanning the same geographical extent. We used Gram-Schmidt pansharpening to combine the multispectral and panchromatic images into one image with strong spatial and spectral resolution.

To train and evaluate the models we manually selected 286 $1000 \times 1000$ pixel slices that, in combination, capture a comprehensive set of visual features representative of the objects of each class across space and time. We trained a team of labelers from the United States and Indonesia to manually label the $1000 \times 1000$ pixel slices, annotating pixels within the

images as belonging to our hypothesized classes of interest: water, foundation, rubble, cloud, beach, agriculture, road, or building. Three labelers were specifically recruited in Aceh, Indonesia, for the task. After an evaluation period, they were hired and paid to work full-time on the project. In particular, they completed all of the labeling used for the testing subset of imagery against which the model was evaluated and segmentation performance was measured. These labelers reside in Aceh and have extensive local knowledge of the study area, including the destruction from the tsunami and subsequent reconstruction.

Each labeler on the team was rigorously trained and thoroughly evaluated to ensure their work was careful and accurate. Each set of labels was carefully reviewed by the U.S.-based study team. This attention to quality control ensured accuracy and consistency across labelers and produced high-quality labelled data.

We worked with a relatively large number of classes for several reasons. First, a priori, we did not know which class or combination of classes would matter for each of the multiple outcomes we consider, and wanted to avoid imposing a "solution" at the outset. Second, from working in the study area in the years after the disaster, we chose classes that corresponded to our field observations over the period in question, and which seemed consistently identifiable in the satellite imagery. Third, although some categories such as road and buildings could be combined in the sense that both are "built infrastructure," they exhibit different patterns in the imagery and we believed we could improve accuracy by labelling and validating them separately, with the option of aggregating at a later stage.

A subset of two hundred $1000 \times 1000$ pixel images was used for training and validation. We reserved the remaining 86 for testing. To accommodate computing constraints while maintaining reasonable minibatch sizes during training, we further sliced each of the $1000 \times 1000$ pixel images in the training and validation subset into sixteen $250 \times 250$ pixel images, resulting in an initial training and validation subset of 3200 images.

To prevent geographic areas or features in the testing subset from overlapping with exact areas or features in the training and validation subset from another point in time, any training or validation image overlapping a testing image at any point in time was removed from the training and validation subset. While images at different points in time may be highly qualitatively distinct due to differences such as destruction or reconstruction, season, time of day, or angle of sensor, the exclusions prevented artificial inflation of generalization performance calculated on the testing subset that might have resulted had we included images more likely to precisely resemble those contained in the training and validation subset because of geographical overlap over time. After removing images overlapping with testing examples, our final training and validation subset consisted of 2780, $250 \times 250$ pixel images. We then split this subset randomly at a 3:1 ratio, yielding a training subset of 2085 images and a validation subset of 695 images.

We also implemented a "hold out one" approach to avoid overlap between locations for which we have STAR survey data on population well-being and areas in the images selected for the training and validation subset. One network model (the "primary network") was trained on all 2085 training images, with its parameters and hyperparameters optimized relative to all 695 validation images. We used this primary network model to characterize areas from which STAR survey data were collected that are not intersected by training or validation examples. For STAR survey areas that overlapped spatially with either training or validation examples, we trained and applied different models that excluded the overlapping data from the training or validation subsets in a hold-one-out approach. We took this step to eliminate the possibility that associations between land cover change and population well-being resulted from overfitting to specific training or validation examples rather than the model's ability to generalize to distinct data in nearby locations. This approach required training 36 total networks, including the primary network. The primary network was used in linking estimates of land cover to the census data, which covers the entire region for which we have imagery.

## Network Architecture and Parameter Initialization

Neural network models organize learnable parameters into layers that sequentially extract features, first from the input data and then from the feature output of each prior layer, to produce predictions informed by abstract, higher-order features.

Our programmatic implementation of DeepLabV3+ is based largely on MATLAB[36]. Training the network requires specifying the network's overall structure and determining optimal values of several model hyperparameters. For the network's encoder module, we use the Xception network with an output stride of 8[17,36,37]. Because our classes are imbalanced, we apply median frequency balancing to upweight lower frequency classes in the loss function, which increases the value of each low frequency class example in the optimization process[38]. We pre-initialize the encoder of the model with ImageNet weights to leverage transfer learning[36,39]. We initialize the decoder section using the Glorot method[40] and retain MATLAB's 10x learn rate in these layers. Seeking to leverage information contained in the near infrared (NIR) band of the QuickBird imagery (this channel is not present in the ImageNet RGB dataset), we perform a simple replacement of the first convolutional layer (which consisted of three-channel convolutional filters), with four-channel filters and initialize the entire layer with the Glorot method[40].

In each of the 36 networks, the remaining hyperparameters (learn rate, momentum, weight decay, minibatch size, and number of training epochs) are determined using 30 iterations of parallel Bayesian hyperparameter optimization (implemented in a manner similar to that of MATLAB D). At each iteration, the Bayesian algorithm selects a set of hyperparameters and trains a complete network end-to-end using stochastic gradient descent with momentum. During training, we shuffle the order of the dataset between epochs, and, as each image is passed through the network, apply several data augmentations: a rotation between −180 and 180 degrees; an X-axis reflection for a random 50% of images; a Y reflection for a random 50% of images; and an X and Y translation each between −15 and 15 pixels for all images.

Unlike stochastic gradient descent, Bayesian hyperparameter optimization builds a global model within the defined hyperparameter search ranges and expends computational time sampling points intelligently. (Supplementary Table 2 provides the specified ranges as well as the optimal values determined for the primary network). As a result the optimization tends to perform well in situations where evaluating each point is computationally expensive, such as training a complete CNN[41,42]. In our work, the Bayesian optimization algorithm maximizes the average of the per-class $F_1$ scores (each the harmonic mean of each class's precision and recall scores) computed on the validation set. We use the "expected improvement plus" acquisition function during the procedure and maintain the default exploration ratio of 0.5[43,44]. As shown in Supplementary Table 2 our hyperparameter optimization scheme permits modification of the baseline DeeplabV3+ network architecture in two significant ways with the potential to improve validation accuracy. First, we permit the substitution of the ReLU nonlinearity with the gaussian error linear unit (GeLU) described in Hendrycks and Gimpel, which demonstrated improved performance in several of their test cases[45]. Additionally, we enable the DeepLabV3+ atrous spatial pyramid pooling (ASPP) module to consider a small and medium set of dilation factors, [1,3,6,9] and [1,6,12,18], respectively, in case a smaller set may be more appropriate for our smaller input image size, in addition to dilation factors which would span a similar number of pixels, [1,12,24,30], relative to the input image as those of the original network from Chen et al., 2018[17].

## Segmentation

Limitations in access to high-memory GPUs necessitate partitioning images into smaller pieces and passing them through a network individually. Because one image slice cannot communicate visual information across the boundary to another, a checkerboard pattern emerges along the edges where constituent images meet once the full landscape is reassembled. Because CNNs train slowly but are relatively quick at inference, a straightforward

solution is to apply the network multiple times, shifted by one third the size of the slice each time along both the vertical and horizontal dimensions. Each segmented image then "votes" in an ensemble, with ties broken randomly. We segment the landscape into images of $250 \times 250$ pixel height and width and then merge the results from the three voters (each shifted by $1/3^{rd}$ vertically and horizontally from the others). The approach reduces the checkerboard pattern once the slices are reassembled into a continuous segmentation because the networks, taken together, generally have access to additional visual information to make the proper classification at points for which such information would otherwise be unavailable due to image boundaries. The confusion matrices presented in Supplementary Tables 3 and 4 compare, for the primary network, test set performance based on segmentation outputs generated without and with, respectively, the benefit of the voting method.

## Computer Vision Results

The $F_1$, Precision, and Recall scores for the primary network optimized on all the training and validation imagery, produced by comparing the 86 labelled, $1000 \times 1000$ pixel images constituting the testing subset to those regions in our large-scale segmentations to incorporate the benefits of the segmentation procedure, are described in Table 4. A full confusion matrix is provided in Supplementary Table 3 for the primary network. Generally, the trained algorithm performs better on more visually homogenous classes such as cloud, beach, and building and the classes with larger quantities of labelled training data such as water and agriculture. Foundations are the least reliably classified (they are also the rarest class with respect to number of pixels), followed by rubble. These lower scores likely arise because of the difficulty of constructing a consistent visual definition of rubble or foundation, the relatively lesser prevalence of each class in the labelled imagery set, and the diversity and irregularity of examples within each class.

Several factors complicate comparing the performance of our trained network to others in the literature. First, the Acehnese landscape consists of urban, suburban, rural, agricultural, shoreline, beach, and forested regions. This diversity is not represented in analyses that focus disproportionately on dense urban settings, as in the analyses of the US and China[46]. Second, our dataset includes challenging post-disaster imagery, which is far less regular than much of the imagery typically used in semantic segmentation of land cover. Third, applications frequently focus only on urban areas and only on one urban class[46]. Fourth, applications to destruction usually consider only a short time frame after the event. We develop a single model that can classify eight classes over eleven time points, analyzing destruction caused by the tsunami simultaneously with reconstruction in its aftermath. Despite these added complexities, our performance is competitive with the literature on the most widely studied classes.

Our primary network achieves a pixel-wise $F_1$ score of 0.8007 segmenting buildings in our satellite imagery testing subset. To take two modern points of comparison, one analysis achieves an 0.809 $F_1$ and another achieves an $F_1$ of 0.7994, each on the urban WHU East Asia Satellite Building Dataset II[47,48]. Our primary network returned an $F_1$ of 0.6050 detecting the diverse paved roads, dirt roads, and pathways of Aceh. Analysis of roads yielded an 0.6494 average $F_1$ for road detection in satellite imagery in the THEOS Thailand road satellite dataset with imagery sampled from five Thai provinces (with per-province $F_1$ scores ranging from 0.550 to 0.775)[49]. An $F_1$ of 0.8154 was achieved with the DeepGlobe Roads Dataset (Thailand, India, Indonesia); 0.7633 on the SpaceNet Roads Dataset (Las Vegas, Paris, Shanghai, Khartoum); and 0.7270 on the CHN6-CUG roads dataset (China)[50]. In a comprehensive paper that segments both buildings and roads in urban Potsdam, Germany[51] the best results (achieved via different models) were an $F_1$ score of 0.913 for buildings and 0.825 for roads.

Most prior work determines damage from natural disasters by locating and classifying the extent of damage to standing buildings, drawing on the xBD dataset, which was created to facilitate the use of satellite imagery for identification of degree of damage after multiple types of hazards[52]. Working with the xBD dataset, Weber and Kan develop recommendations for image processing and training procedures that achieve a localization (whether an object is background or building) $F_1$ of 0.835 and a damage detection $F_1$ (extent of damage to the building) of 0.697 within a single network[53]. For southern Turkey reported precision and recall generate an $F_1$ of 0.8226 in work identifying damaged buildings (the authors link their satellite method with data on the distribution of the population to provide information regarding the relative urgency for rescue efforts across locations)[54]. Sodeinde et al. evaluate the xBD dataset across hazard types[55]. Their work suggests that under-representation of imagery from earthquakes, volcanoes, and tsunamis substantially reduces the model's utility for detecting and classifying building damage from these types of hazards[55]. Similar methods have been applied to building damage from other causes. For example, in Syria, where building destruction has resulted from a civil war, an average precision (across six cities) of about 0.42 is achieved in detecting destruction from satellite imagery using machine learning tools[56].

Efficient and accurate landcover classification with remotely sensed imagery has long confronted challenges resulting from high-dimensional data, variations across sensors, limited labeled samples, and difficulties in leveraging high performance in one region into high performance in another, among other challenges[7]. Deep learning methods have effectively identified and confronted some of these and other difficulties[57], and while landcover analysis in very-high resolution, remotely-sensed imagery remains challenging, semantic segmentation methods may be able to ultimately overcome these challenges in combination with other techniques, including domain adaptation or transfer learning, and with improvements in the availability of training data and the implementation of training strategies designed to efficiently leverage sparse examples[58]. Nonetheless, a trend towards achieving more general and global-level classification rather than high, but very specific, accuracy with more complicated deep learning models may have begun to emerge in light of insufficiencies in the quantity and quality of available data and to leverage sensors with different modalities[58].

Large vision foundation models have expanded the generalizability of machine learning tools in video and imagery applications, and, once trained, can perform well even in the[59,60] zero-shot context, but the success of these models applied directly to remote sensing tasks has been limited, even in the context of RGB images, in part because of the differences in domain between typical, widely available imagery data and remote sensing imagery data[60,61]. While fine-tuning a general application foundation model with remote sensing data would be expected to improve domain-specific performance, the presently available remote sensing datasets may not be sufficiently large or of sufficient quality to train a true foundation model capable of high-

## Table 4 | Segmentation Evaluation

| Metric | Agriculture | Water | Building | Other vs. Main 3 | Road | Foundation | Rubble | Beach | Cloud | Other vs. All |
|---|---|---|---|---|---|---|---|---|---|---|
| F1 | 0.7554 | 0.7594 | 0.8007 | 0.8874 | 0.6050 | 0.2227 | 0.3935 | 0.8174 | 0.9090 | 0.8244 |
| Precision | 0.6950 | 0.7546 | 0.7096 | 0.9351 | 0.5163 | 0.1823 | 0.4492 | 0.7582 | 0.9645 | 0.8889 |
| Recall | 0.8271 | 0.7644 | 0.9185 | 0.8443 | 0.7304 | 0.2862 | 0.3500 | 0.8866 | 0.8596 | 0.7686 |

Per-class and arithmetic means of Precision, Recall, and F1 scores for the primary network optimized with the complete training and validation sets. The evaluation of the network on the $1000 \times 1000$ test set slices as actually present in our large-scale segmentations (which benefit from the voting procedure).

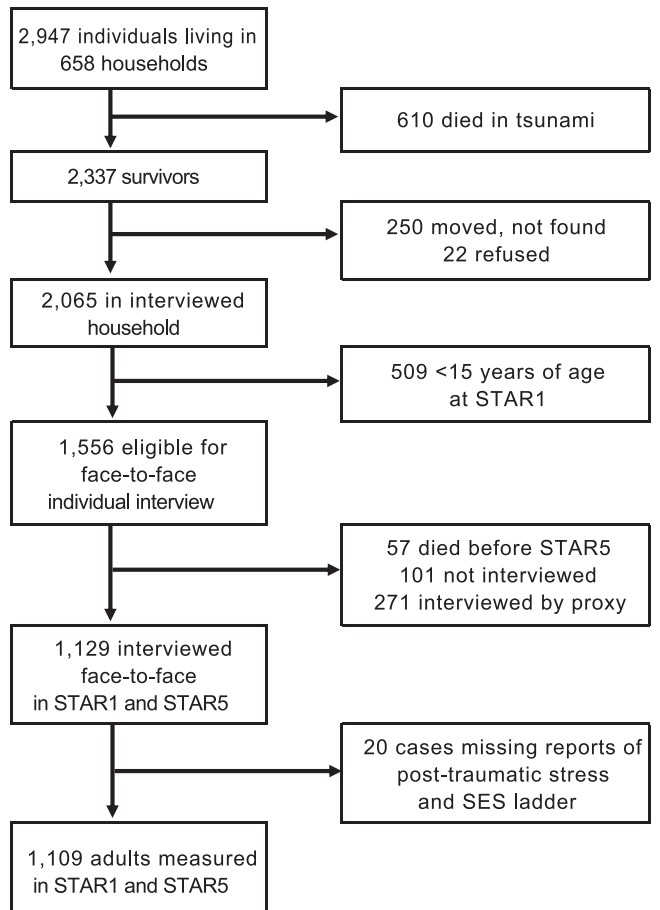

**Fig. 6 | Survey outcomes.** Flowchart of individuals who were eligible to participate in the 2005 and 2009 STAR surveys because they were members of households located (in 2004) in one of the 43 survey enumeration areas for which high resolution satellite imagery is available of four points in time. Of the 2,947 eligible individuals 610 died in the tsunami. Others could not be interviewed after the tsunami or died between the 2005 and 2009 interview.

quality zero-shot predictions[61]. There is, however, growing interest in the application of foundation models to remote sensing, which could ultimately reduce human effort for dataset collection and annotation, facilitate transfer learning and fine-tuning, and expand access to machine learning tools[60,62].

In our primary network, most classes achieve higher recall (the proportion of objects of a class in the test dataset correctly identified by the model) than precision (the proportion of model predictions of a class that match the manually assigned test set label), implying generally that the network's predictions regarding what belongs to each class reliably captured the members labelled as belonging to that class but the network *over-predicted* members of that class, and included examples which the labels confirmed belonged to other classes.

## Using the Segmentation Datasets

The network produces two-dimensional matrices of data of equivalent size to the underlying satellite imagery, in which each cell contains the class assignment for the corresponding pixel. The matrices are georectified relative to the underlying imagery using ArcGIS, after which each pixel corresponds to a known latitude and longitude. Information can be aggregated across pixels to characterize land cover for areas of particular sizes, for administrative areas, or for zones with particular attributes, as in Figs. 4 and 5. We characterize an area of 287 km², for which imagery is available for four points in time. This area contains 164 administrative units falling within the city of Banda Aceh and the surrounding district of Aceh Besar and 43 STAR enumeration areas. We link census data for the 164

administrative units to segmentations produced by the primary network. We link STAR data to the segmentations from the primary network for the survey EAs that did not overlap a training or validation example, whereas EAs with overlap are linked to the 35 EA-specific networks.

## Data on Population

We draw on population data from two censuses, conducted in 2005 and in 2010 by Statistics Indonesia, which provide counts of the population for each of the 164 administrative areas. Our other data source is the Study of the Tsunami Aftermath and Recovery (STAR). STAR is a longitudinal survey of individuals, households and communities. The baseline was conducted 10 months before the tsunami as part of the 2004 National Socioeconomic Survey (SUSENAS), a survey conducted annually by Statistics Indonesia that is representative of the population at the district (*kabupaten*) level[63]. The STAR survey targeted the SUSENAS enumeration areas in thirteen districts in Aceh and North Sumatra, all with coastlines potentially vulnerable to inundation from the tsunami. The STAR sites capture variation in disaster severity. Tsunami survivors and their children born after the tsunami were tracked and interviewed annually for 5 years after the disaster and again in a 10-year follow-up in 2015 and in a 15 year follow up in 2020[20]. Attrition rates in all waves are low, resulting from the survey team's persistence in tracking respondents. Of the 26,919 individuals who were members of baseline households in 2004, 24,752 survived the tsunami. Follow-up rates with the individual panel exceed 90% in post-tsunami waves. In each survey we interviewed panel respondents and new members of households. The project received approval from the University of North Carolina IRB (protocol number 20-2673).

For this paper we focus on the subset of STAR respondents who, in 2004, were living in an enumeration area (EA) within the area for which we have satellite imagery at four points in time. There are 43 EAs for which imagery data is available. In 2004, before the tsunami, these EAs were home to 658 households (containing 2,947 individuals) that were randomly selected for participation in the 2004 SUSENAS. Survey outcomes are depicted in Fig. 6.

STAR collects detailed data at the household level on household composition, including the survival status of each of the 2004 household members and where they currently live if they are alive but no longer residing in their original household. Individuals are interviewed about a range of topics, including questions about physical and psychosocial health and perceptions of socioeconomic status before the tsunami and at the interview.

We use the STAR data to construct measures of mortality between the 2004 and 2005 interview for each member of a 2004 household, and of displacement at the 2005 interview for those who survived the tsunami. We also construct a measure of intensity of symptoms for post-traumatic stress, based on seven items from the Post-traumatic Stress Checklist (PCL, civilian version), which has been widely validated[64]. Lastly we draw on respondents' answers to questions asking them to rate their position on a 6-step ladder of socioeconomic status[65].

## Analysis and Statistics

To relate land cover to population dynamics we present summary statistics and results from regressions in Tables 2 and 3. We estimate an OLS regression for the log of an administrative area's population size in 2005 as a function of land cover shares in 2004 and 2005. We also analyze the log of the mean annualized population growth rate between 2005 and 2010 (a community-level outcome, computed as $[\ln(P_{10}/P_{05})/5]$). In these regressions standard errors are adjusted for spatial autocorrelation using a linear decay function that declines to 0 at distances of 7 km from the community center[66,67].

To examine outcomes for individuals in 2005, measured in the STAR data, we estimate the following multivariate regression:

$$\theta_{i05} = \alpha + \beta S_{c05} + \gamma S_{c04} + \delta X_{i04} + \varepsilon_i \qquad (1)$$

where $\theta_{i05}$ is an outcome measure for individual i in 2005, $S_{c05}$ and $S_{c04}$ are vectors of land use shares in 2005 and in 2004 for the community c in which individual i resided at the time of the tsunami, $X_i$ is a vector of individual characteristics (age, sex, educational level, and district of residence, measured in 2004), and $\varepsilon_i$ represents unobserved heterogeneity. All test statistics are calculated taking into account clustering at the EA levels.

Individual-level outcome measures are death in the tsunami, displacement in the first four months after the tsunami, post-traumatic stress reactivity (PTSR) score, and the difference between an individual's perception of SES after the tsunami relative to before the tsunami, as reported in the first post-tsunami interview. Mortality and displacement are measured with dichotomous variables that take on a value of 1 if an individual perished or was displaced, 0 otherwise). PTSR varies from 0 to 21, change in the SES ladder from before to shortly after the tsunami varies from −6 to 6.

We also estimate individual-level regressions in which the outcomes are change over time:

$$(\theta_{i09} - \theta_{i05}) = \alpha + \beta(S_{ci09} - S_{ci05}) + \delta X_{i04} + \varepsilon_i \qquad (2)$$

Where ($\theta_{i09}$ - $\theta_{i05}$) is the difference between a measure of well-being in 2009 and that measure in 2005. In this specification ($S_{ci09}$-$S_{ci05}$) is a vector of changes over time in the shares of land cover accounted for by water, buildings, and agriculture. As before, $X_i$ is a vector of individual characteristics (age, sex, educational level, and district of residence, measured in 2004), and $\varepsilon_i$ represents unobserved heterogeneity. We consider two individual-level outcomes: change in level of PTSR between 2005 and 2009 and change in position on the SES ladder between 2005 and 2009. We perform two-tailed tests to assess statistical significance of regression coefficients.

## Reporting summary
Further information on research design is available in the Nature Portfolio Reporting Summary linked to this article.

## Data availability
The population data we used are from the Study of the Tsunami Aftermath and Recovery (STAR), which builds from baseline data that were part of the 2004 National Socioeconomic Survey (SUSENAS) conducted by Statistics Indonesia, and from censuses conducted by Statistics Indonesia. Statistics Indonesia policy restricts access to data in which respondents' neighborhoods are identifiable, which precludes the release of the analytical datasets on which population results are based. The STAR survey data, without neighborhood identifiers, are available at star-data.org. Statistics Indonesia provides access to its surveys and censuses through its portal (https://www.bps.go.id). We are unable to redistribute the satellite imagery used to train the models due to data licensing restrictions of Digital Globe (now https://www.maxar.com) who provided the satellite imagery.

## Code availability
The maps and linkage of segmentation matrices to underlying imagery were produced using ArcGIS. The CNN model is implemented with DeepLabV3+ using MATLAB. The statistical analysis relating segmentations to population outcomes was performed in STATA 19.

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

## Acknowledgements

We acknowledge funding from the Eunice Kennedy Shriver National Institute of Child Health and Human Development (R21051970, R01HD052762, P2C HD050924), National Institute on Aging (R01AG031266, R01AG065395), and the National Science Foundation (CMS-0527763). The content is solely the responsibility of the authors and does not necessarily represent the official views of the National Institutes of Health or other funding agencies. We are grateful to the Digital Globe Foundation for providing the satellite images we analyze through its imagery grant program and to Thomas Gillespie and three anonymous reviewers for their comments.

## Author contributions

E.P. wrote the machine learning code. E.P. and E.F. prepared the manuscript with input from all the authors. E.P. and P.K. processed the images, E.P. and C.S. developed the labelling protocols, P.K. linked the land cover measures to maps with administrative boundaries. E.F. and D.T. analyzed the census and survey data. E.P., E.F., C.S., D.T. interpreted the results.

## Competing interests

The authors declare no competing interests.
