## [Transparent Peer Review file · Communications Earth & Environment]

High-resolution imagery and neural networks link post-tsunami land cover changes to population health and well-being

Corresponding Author: Professor Elizabeth Frankenberg

Version 0:

Decision Letter:

Dear Professor Frankenberg,

First of all, please allow me to apologise for the delay in sending a decision on your manuscript titled "Using High Resolution Imagery and Neural Networks to Measure Destruction and Reconstruction after a Disaster". It has now been seen by 3 reviewers, whose comments are appended below. You will see that they find your work of some potential interest. However, they have raised substantial concerns that must be addressed. In light of these comments, extensive revisions will be required before we can further consider the manuscript for publication. We would, however, be interested in considering a revised version that fully addresses these serious concerns.

We hope you will find the reviewers' comments useful as you decide how to proceed. Should additional work allow you to

- address these criticisms (that is, either to incorporate the suggestions or provide a compelling argument why the point made by the reviewer is not valid or relevant to the editorial threshold as outlined below)

AND

- meet our editorial thresholds as outlined below,

then we would be happy to look at a revised manuscript.

In the following, we list our requirements for publication.

- Present novel and fully supported insights into how changes in socio-economic status can lead to landcover change following the 2004 Indian Ocean tsunami.

- Provide an overview of the literature on land classification models.

- Justify your method in detail, including sample selection and the selection of the 2004 Indian Ocean tsunami, and clarify the validation process.

- Discuss the limitations of your findings and approach, their potential use in future disasters and relief efforts, and their overall policy implications.

If you choose to take up this option, please either highlight all changes in the manuscript text file, or provide a list of the changes to the manuscript with your responses to the reviewers.

When resubmitting, please provide a point-by-point response to the reviewers' comments. Please submit your responses as a separate file, distinct from your cover letter where you can add responses to the Editors' comments that you do not want to be made available to the reviewers. Word files are preferred. We recommend that any figures, tables or graphs that are included in the response to reviewers are also included in the main article or Supplementary Information.

If the revision process takes significantly longer than three months, we will be happy to reconsider your paper at a later date, as long as nothing similar has been accepted for publication at Communications Earth & Environment or published elsewhere in the meantime.

Please use the following link to submit your revised manuscript, point-by-point response to the reviewers' comments with a list of your changes to the manuscript text (which should be in a separate document to any cover letter), a tracked-changes version of the manuscript (as a PDF file) and any completed checklist:

Link Redacted

Please do not hesitate to contact us if you have any questions or would like to discuss the required revisions further. Thank you for the opportunity to review your work.

Best regards,

C. Kendra Gotangco Gonzales, PhD
Editorial Board Member
Communications Earth & Environment
orcid.org/0000-0002-3436-9813

Martina Grecequet, PhD
Senior Editor
Communications Earth & Environment
Consulting Editor
Communications Sustainability

EDITORIAL POLICIES AND FORMAT

If you decide to resubmit your paper, please ensure that your manuscript complies with our editorial policies and complete and upload the checklist below as a Related Manuscript file type with the revised article:

- Behavioural and social science
- Ecological, evolutionary & environmental sciences
- Life sciences

For your information, you can find some guidance regarding format requirements summarized on the following checklist: (<https://www.nature.com/documents/commsj-phys-style-formatting-checklist-article.pdf>) and formatting guide (<https://www.nature.com/documents/commsj-phys-style-formatting-guide-accept.pdf>).

REVIEWER COMMENTS:

Reviewer #2 (Remarks to the Author):

1- Why was the 0.3km² at the geographic center of each administrative unit chosen for the admin unit sample? The centroid is not necessarily particularly meaningful depending on the size/density/etc of the admin unit. This may be sufficiently large to cover most of the unit since it is sub city/district, but given the impact this choice has on the data utilized in the analysis, more clarification/justification is needed. For example, a statistic along the lines of what the average percentage of each admin unit the 0.3km² covers would be informative. If the 0.3km² covers a relatively small proportion of many of the units, I would be concerned about the characteristics of the area selected vs the overall admin unit, and the broader implications on the analysis.

2- I appreciate the value in using a broader set of classes for training your models, but given the ultimate focus on the three more traditional classes (buildings, ag, water) there is a significant amount of related literature (existing labeled data, pre trained models, etc). You touch on this a bit in the discussion around the CV model, but there is sufficient literature on generalizable land cover classification models, as well as a growing body based on foundation models, that it is worth discussing in some more detail and with references. I don't think it is necessary for you to incorporate / adapt your approach or analysis for this paper, but I do think this broader body of work (outside disaster applications) should be considered in the context of future work, etc. Since you are already leveraging transfer learning, it would be very reasonable to use existing land classification training data (or trained models) as an additional layer of model fine tuning before training on your data. It may also be worth providing more details on how the additional classes specific to the study area / disaster applications facilitated your validation process (particularly considering the lower performance metrics of foundations/rubble).

3- I think this paper would benefit from building on the current discussion section to consider how these findings and approaches could be leveraged to support populations impacted by future disasters. The current methodological approach and findings linking land cover change to human well being are meaningful, but connecting these more explicitly with policy implications and potential mechanisms to support disaster relief efforts would be a strong addition to round out this piece. E.g., post disaster recovery trends based on shifts in land cover could be used to redirect aid to areas where recovery is lagging.

Reviewer #3 (Remarks to the Author):

Review COMMSENV-25-2581-T "Using High Resolution Imagery and Neural Networks to Measure Destruction and Reconstruction after a Disaster" Frankenberg et al.

The purpose of the paper is to relate demographic and socioeconomic data to landcover changes over a ca. 5 year period extracted with CNN from multitemporal Quickbird images of the Banda Aceh area in Indonesia. The authors intend to investigate to what extent changes in well-being and socioeconomic status can be linked to the specific destruction caused by the 2004 tsunami and subsequent recovery, as expressed by changes in 8 landcover classes. The paper is positioned as a proof of concept.

I do not share the view that this is pioneering work where a new way of linking satellite remote sensing and field-collected data is shown. Specific comments:

- The time period under investigation here is ca. 2004-2010. Given the large amount of research the 2004 tsunami disaster received, especially in the most heavily affected area of northern Sumatra, information given on the specific recovery process of the households investigated is simply of little wider interest, more than 20 years after the event. The study also draws heavily on results of the longitudinal Study of the Tsunami Aftermath and Recovery (STAR) project, which has already featured in numerous publications, including several by the first author of this manuscript

- The image analysis part is presented as particularly novel. However, while the CNN- analysis is competently executed, using machine learning-based remote sensing data analysis in post-disaster recovery assessment, including over multiple time steps, is not new – see for example the work by Mohammadreza Sheykhmousa or Saman Ghaffarian. The latter in particular used deep learning, but also Google Earth Engine and Digital Twins.

- I am also not convinced of the analysis results. The Quickbird images used are described as high resolution 60 cm images, and only in the Supplementary Information it is described that the data are 2.4 m multispectral and 61 cm panchromatic, the latter used to pansharpener the former – this means that, while at the time of the disaster these data were indeed novel and considered high-resolution, their level of detail not really sufficient to map specific classes such as foundations, buildings and rubble, let alone what are actually landuse classes (agriculture for example). The limitations of Quickbird images have become clear in other disasters around this time, such as the 2006 Bantul earthquake.

- Another problem is that in this study a tsunami is equated with a flood (L49, "We focus on flooding". Sure, both involve water, but they are as different as a storm from a tornado in terms of destructiveness and the resulting damage picture. Tsunamis, and also storm surges (see damage and recovery analysis of post-Haiyan Tacloban), do not simply damage buildings, but create continuous debris fields with floating debris filling most of the space between buildings, often pushing material into and onto building parts. Separating actual buildings, and identifying their damage state, within such a mess is nearly impossible, even in highly detailed drone images. Quickbird cannot reliably reveal actual damage, which is why I have little confidence in the analysis results shown in Fig 2 (panels for 2004DEC30 and 2005AUG06) - likely the actual amount of destroyed buildings is lower.

- A final point of doubt concerns the damage labelling by volunteers. As someone who was closely involved in post-2010 Haiti GEOCAN effort and other crowdsourced damage labelling efforts I do not believe that it can be done reliably, especially in what are still coarse satellite images (for Haiti 1.24m resolution Geosy 2 images were used for relatively simple seismic damage, and it still did not work). But when using those results in the CNN those errors just propagate. Given the many issues I have with the manuscript I cannot recommend publication

Reviewer #4 (Remarks to the Author):

The authors choose the 2004 Indian Ocean tsunami as a test case for flooding damage and hazards. I feel that this is not a suitable choice. First, this tsunami was induced by an earthquake, which is not related to climate change and sea level rise as indicated in the introduction section. Second, the type and nature of damage after tsunami would be very different

compared to the more typical flooding events due to weather such as extreme rainfall and storm surges. The damage due to tsunami is much more severe and extensive and thus is easily mapped. In contrast, the damage is more subtle in typical flooding events, that would require a more refined algorithm for detection. The CNN model was trained for tsunami damage. It is not obvious whether it would work for typical flooding events. The methodology and results are sound for this tsunami event. The method is good for a study of damage and recovery after a disaster such as the destructive tsunami in this case. However I would not recommend it for typical flooding events. I would like to see use cases in more typical flooding events that are becoming the norms nowadays.

** Visit Nature Portfolio's author and referees' website at www.nature.com/authors for information about policies, services and author benefits**

Communications Earth & Environment is committed to improving transparency in authorship. As part of our efforts in this direction, we are now requesting that all authors identified as 'corresponding author' create and link their Open Researcher and Contributor Identifier (ORCID) with their account on the Manuscript Tracking System prior to acceptance. ORCID helps the scientific community achieve unambiguous attribution of all scholarly contributions. You can create and link your ORCID from the home page of the Manuscript Tracking System by clicking on 'Modify my Springer Nature account' and following the instructions in the link below. Please also inform all co-authors that they can add their ORCIDs to their accounts and that they must do so prior to acceptance.
<https://www.springernature.com/gp/researchers/orcid/orcid-for-nature-research>

Version 1:

Decision Letter:

Dear Professor Frankenberg,

Your manuscript titled "Using High Resolution Imagery and Neural Networks to Measure Destruction and Reconstruction after a Disaster" has now been seen by our reviewers, whose comments appear below. In light of their advice we are delighted to say that we are happy, in principle, to publish a suitably revised version in Communications Earth & Environment.

We therefore invite you to revise your paper one last time to address the remaining concerns of our reviewers. At the same time we ask that you edit your manuscript to comply with our format requirements and to maximise the accessibility and therefore the impact of your work.

EDITORIAL REQUESTS:

****Please take care to match our formatting and policy requirements. We will check revised manuscript and return manuscripts that do not comply. Such requests will lead to delays. ****

SUBMISSION INFORMATION:

OPEN ACCESS:

Communications Earth & Environment is a fully open access journal. Articles are made freely accessible on publication. For further information about article processing charges, open access funding, and advice and support from Nature Portfolio, please visit <https://www.nature.com/commsenv/open-access>

Link Redacted

Best regards,

Charlotte Kendra Gotangco Gonzales, PhD
Editorial Board Member
Communications Earth & Environment

Martina Grecequet, PhD
Senior Editor,
Communications Earth & Environment
Consulting Editor,
Communications Sustainability

REVIEWERS' COMMENTS:

Reviewer #2 (Remarks to the Author):

I found the responses and associated revisions to my feedback largely satisfactory from a technical perspective. While they did reasonably address my final comment about connecting their work and findings to practical policy considerations, I still feel like this may be an underdeveloped aspect of the paper. Although other reviewers raised more explicit concerns regarding the novelty of the work based on the methods, I felt the adaptation of these methods to a relevant and novel use case had the potential to balance out such concerns. That said, a critical element of working with a 20+ year old use case is connecting it with current efforts around disaster response. Beyond the authors providing additional suggestions regarding the use of their methods, I am not sure how they can reasonably address this further. Ultimately, I find the work to be sound but leave it up to the Editor to consider these concerns and whether the paper in its current form justifies publication in this journal.

Reviewer #4 (Remarks to the Author):

The revised manuscript has adequately addressed the concerns in the first review. I have no further comments.

** Visit Nature Portfolio's author and referees' website at <http://www.nature.com/authors> for information about policies, services and author benefits**

POINT BY POINT RESPONSE TO THE REVIEWERS

Reviewer #2 (Remarks to the Author):

- 1- Why was the 0.3km² at the geographic center of each administrative unit chosen for the admin unit sample? The centroid is not necessarily particularly meaningful depending on the size/density/etc of the admin unit. This may be sufficiently large to cover most of the unit since it is sub city/district, but given the impact this choice has on the data utilized in the analysis, more clarification/justification is needed. For example, a statistic along the lines of what the average percentage of each admin unit the 0.3km² covers would be informative. If the 0.3km² covers a relatively small proportion of many of the units, I would be concerned about the characteristics of the area selected vs the overall admin unit, and the broader implications on the analysis.**

We are grateful to the reviewer for raising this good point regarding the use of the centroid in the analysis of the administrative sample unit. In response to the concern, we have revised our approach and constructed land cover statistics for each administrative unit that characterize the entire geographic expanse of that unit. As a result, our measures better capture the changes within each unit, during and after the disaster.

Some administrative units in our study area share borders which has no impact on coefficient estimates but may affect the precision of those estimates. We allow for spatial correlation in unobserved characteristics in the models, following Cameron, Gelbach and Miller (2011) with refinements suggested by Colella et al. (2019, 2023). Specifically, we allow geographic-based correlations both within and across administrative units and allow the degree of correlation to decay with increasing distance from the center of the administrative unit. After experimenting with different distance cut-offs, we selected a distance of 7kms because standard errors were largest at that distance and, therefore, our inferences are conservative. It turns out that none of our conclusions regarding significance of estimates is affected by the choice of the distance cut-off.

We discuss this revised approach on page 3, with reference to Figures 1, 4, and 5 and Table 1.

- 2- I appreciate the value in using a broader set of classes for training your models, but given the ultimate focus on the three more traditional classes (buildings, ag, water) there is a significant amount of related literature (existing labeled data, pre trained models, etc). You touch on this a bit in the discussion around the CV model, but there is sufficient literature on generalizable land cover classification models, as well as a growing body based on foundation models, that it is worth discussing in some more detail and with references. I don't think it is necessary for you to incorporate / adapt your approach or analysis for this paper, but I do think this broader body of work (outside disaster applications) should be considered in the context of future work, etc.**

Thank you for this suggestion, we have added a broader discussion of literature on generalizable land classification models and foundation models. This section of text is reproduced below, and appears on page 11.

Efficient and accurate landcover classification with remotely sensed imagery has long confronted challenges resulting from high dimensional data, variations across sensors, limited labeled samples, and difficulties in leveraging high performance in one region into high performance in another, among other challenges (Zhao et al. 2023). Deep learning methods have effectively identified and confronted some of these and other difficulties (Li et al. 2024), and while landcover analysis in very-high resolution, remotely-sensed imagery remains challenging, semantic segmentation methods may be able to ultimately overcome these challenges in combination with other techniques, including domain adaptation or transfer learning, and with improvements in the availability of training data and the implementation of training strategies designed to efficiently leverage sparse examples (Qin and Liu 2022). Nonetheless, a trend towards achieving more general and global-level classification rather than high, but very specific, accuracy with more complicated deep learning models may have begun to emerge in light of insufficiencies in the quantity and quality of available data and to leverage sensors with different modalities (Qin and Liu 2022).

Large vision foundation models have expanded the generalizability of machine learning tools in video and imagery applications, and, once trained, can perform well even in the zero-shot context, but the success of these models applied directly to remote sensing tasks has been limited, even in the context of RGB images, in part because of the differences in domain between typical, widely available imagery data and remote sensing imagery data (Xiao et al. 2025; Huo et al., 2025). While fine-tuning a general application foundation model with remote sensing data would be expected to improve domain-specific performance, the presently available remote sensing datasets may not be sufficiently large or of sufficient quality to train a true foundation model capable of high-quality zero-shot predictions (Xiao et al. 2025). There is, however, growing interest in the application of foundation models to remote sensing, which could ultimately reduce human effort for dataset collection and annotation, facilitate transfer learning and fine-tuning, and expand access to machine learning tools (Huo et al., 2025; Strong et al. 2025).

The new references are as follows:

Huo C, Chen K, Zhang S, Wang Z, Yan H, Shen J, Hong Y, Qi G, Fang H, Wang Z. When Remote Sensing Meets Foundation Model: A Survey and Beyond. *Remote Sensing*. 2025; 17(2):179. <https://doi.org/10.3390/rs17020179>

Strong B, Boyda E, Kruse C, Ingold T and Maron M (2025) Digital applications unlock remote sensing AI foundation models for scalable environmental monitoring. *Front. Clim.* 7:1520242. doi: 10.3389/fclim.2025.1520242

Zhao S, Tu K, Ye S, Tang H, Hu Y, Xie C. Land Use and Land Cover Classification Meets Deep Learning: A Review. *Sensors (Basel)*. 2023 Nov 3;23(21):8966. doi: 10.3390/s23218966. PMID: 37960665; PMCID: PMC10649958.

Li, Ziming, Bin Chen, Shengbiao Wu, Mo Su, Jing M. Chen, Bing Xu, Deep learning for urban land use category classification: A review and experimental assessment. *Remote Sensing of Environment*, Volume 311, 2024, 114290, ISSN 0034-4257, [://doi.org/10.1016/j.rse.2024.114290](https://doi.org/10.1016/j.rse.2024.114290).

Qin, R. and Tao Liu. 2022. A Review of Landcover Classification with Very-High Resolution Remotely Sensed Optical Images—Analysis Unit, Model Scalability and Transferability" (2022) - *Remote Sensing*

Xiao, A. et al. Foundation Models for Remote Sensing and Earth Observation. *IEEE Geoscience and Remote Sensing Magazine*. 2025.

Since you are already leveraging transfer learning, it would be very reasonable to use existing land classification training data (or trained models) as an additional layer of model fine tuning before training on your data. It may also be worth providing more details on how the additional classes specific to the study area / disaster applications facilitated your validation process (particularly considering the lower performance metrics of foundations/rubble).

We appreciate the suggestion about adding an existing land classification trained model or data as a way to fine tune the model and plan to experiment with this addition in our next analysis phase, at which point we also hope to have obtained more imagery that will let us extend our analysis beyond 2009.

Thank you also for the suggestion about more discussion of the additional classes. We considered 8 classes for the reasons described below and have added this discussion on page 8.

We worked with a relatively large number of classes for several reasons. First, *a priori*, we did not know which class or combination of classes would matter for each of the multiple outcomes we consider, and wanted to avoid imposing a “solution” at the outset. Second, from working in the study area in the years after the disaster, we chose classes that corresponded to our field observations over the period in question, and which seemed consistently identifiable in the satellite imagery. Third, although some categories such as road and buildings could be combined in the sense that both are “built infrastructure,” they exhibit different patterns in the imagery and we believed we could improve accuracy by labelling and validating them separately, with the option of aggregating at a later stage.

- 3- I think this paper would benefit from building on the current discussion section to consider how these findings and approaches could be leveraged to support populations impacted by future disasters. The current methodological approach and findings linking land cover change to human well being are meaningful, but connecting these more explicitly with policy implications and potential mechanisms to support disaster relief efforts would be a strong addition to round out this piece. E.g., post disaster recovery trends based on shifts in land cover could be used to redirect aid to areas where recovery is lagging.**

This is an important point. We now introduce this topic in the Introduction and have added material in the Discussion. The discussion section now concludes with new paragraphs, reproduced below:

A contribution of this study is that it provides a real-world illustration of the value of linking measures of damage to the built and natural environment to high-quality measures of the experiences of exposed populations both immediately after the disaster and over the longer-term. This is important because immediate impacts differ from longer-term impacts and those differences provide insights into the population groups that are most vulnerable in the aftermath of a disaster. Although some work addresses recovery by characterizing trajectories of changes in landcover over time for landscapes affected by disasters almost no other work directly links measures of variation in land cover to measures of variation in population well-being or considers how these relationships evolve over multiple years as reconstruction efforts take place. Expanding attention to population well-being and later phases of the post-disaster period is fundamental if the methods are to realize their full potential.

These insights are important for science and policy. They have the potential to inform real-time evaluations of aid programs while they are being delivered by, for example, documenting the actual timeline of rebuilding of roads, bridges, and housing, re-emergence of agricultural cultivation and other sources of livelihoods. The data could guide dynamic allocation of assistance, ensuring that investments are effective and support the affected populations. Merging information on changes in land cover and land use with data on population measures of socioeconomic status and well-being provides a far richer and more comprehensive picture of immediate impacts of a major disaster and the longer-term trajectories of well-being than does either source alone. The resulting data are key for the design and evaluation of assistance programs, identifying communities in greatest need as recovery efforts are rolled out, and the impacts of those programs.

Reviewer #3 (Remarks to the Author):

Review COMMSENV-25-2581-T “Using High Resolution Imagery and Neural Networks to Measure Destruction and Reconstruction after a Disaster” Frankenberg et al.

The purpose of the paper is to relate demographic and socioeconomic data to landcover changes over a ca. 5 year period extracted with CNN from multitemporal Quickbird images of the Banda Aceh area in Indonesia. The authors intend to investigate to what extent changes in well-being and socioeconomic status can be linked to the specific destruction caused by the 2004 tsunami and subsequent recovery, as expressed by changes in 8 landcover classes. The paper is positioned as a proof of concept.

I do not share the view that this is pioneering work where a new way of linking satellite remote sensing and field-collected data is shown.

The contribution of this work lies not in providing a new way of linking data but in establishing the predictive value of measures of land cover and land use constructed from readily-available satellite imagery for understanding the immediate and longer-term impacts of a disaster on the well-being of human populations. We establish that the measures of destruction and reconstruction based on satellite imagery are predictive of multiple different measures of well-being at the individual and community levels. These include mortality, displacement, mental health and economic status at the individual level and population size and growth at the community level. The immediately post disaster patterns that we document have not been established in the literature. In addition, a pioneering feature of the methods in this study is establishing that changes in landcover and landuse over a period of four years post-disaster are predictive of changes in population well-being over the same time period. We believe the results lay the foundation for studies that assess impacts of disasters, evaluate programs to aid those affected by disasters and to allocate resources to those in greatest need both immediately after the disaster and in the ensuing years.

Although others have used imagery to measure damage after disasters, relatively little work has rigorously assessed accuracy of, for example, damage maps, because of “lack of systematically acquired ground information” (Kerle 2010). We establish the relevance of these types of measures through their ability to predict a series of well-measured critically important outcomes for humans. We have clarified our description of our contributions to emphasize that it is not the way the measures are linked, which is straightforward, but what the measures reveal about well-being and recovery for human populations.

Specific comments:

- The time period under investigation here is ca. 2004-2010. Given the large amount of research the 2004 tsunami disaster received, especially in the most heavily affected area of northern Sumatra, information given on the specific recovery process of the households investigated is simply of little wider interest, more than 20 years after the event. The study also draws heavily on results of the longitudinal Study of the Tsunami Aftermath and Recovery (STAR) project, which has already featured in numerous publications, including several by the first author of this manuscript

A key innovation in this paper and important contribution to science is combining output from our CNN regarding damage sustained by a large-scale natural disaster with high-quality, fine-grained individual- and household-level measures of human suffering and loss. This is the first study to successfully integrate these two distinct literatures and, in so doing, provides a roadmap for future studies of the impacts of disasters. Our social science data come from censuses conducted by the Government of Indonesia and from the STAR survey that we designed and fielded. The STAR survey collects data on a wide range of characteristics and outcomes. No previous publication using STAR data relates the outcomes we consider here to detailed measures of land cover change based on high-resolution satellite imagery before and over four years post-tsunami.

- The image analysis part is presented as particularly novel. However, while the CNN- analysis is competently executed, using machine learning-based remote sensing data analysis in post-disaster recovery assessment, including over multiple time steps, is not new – see for

example the work by Mohammadreza Sheykhmousa or Saman Ghaffarian. The latter in particular used deep learning, but also Google Earth Engine and Digital Twins.

We appreciate the references to the important work of Ghaffarian and Sheykhmousa. We have identified several relevant papers that focus on recovery, or cutting edge approaches to disaster management that involve digital twins, and added references to them in the revised paper. These include Sheykhmousa et al. (2019); Ghaffarian, (2025); Ghaffarian and Lagap (2025).

Sheykhmousa et al (2019) makes several points that underscore the value-added of our study. They note that recovery is the “least understood phase of disaster management” and that many social audit studies are limited. They also comment on the potential of remote sensing methods. We agree that remote sensing studies have been underutilized in studies of recovery. This was one of the motivations underlying our work. We believe that the combination of remote sensing with high quality censuses and surveys provides far more potential for understanding how disasters change lives than either method alone. We note that the studies using remote sensing to assess recovery focus mainly on the nature of landcover changes as it shifts from patterns that suggest damage and debris to patterns those that suggest land use patterns that existed before the disaster. A major contribution of our approach is that we have placed changes in population well-being and economic resource levels of individuals at the center of the transition after a disaster. Ghaffarian (2025) notes the relevance of social and economic factors, as well as field surveys, for disaster risk management, which are the types of sources we analyze. As noted in response to points 1 and 2 above, we believe that our primary contribution to the literature lies in the combination of CNN analysis with detailed high-quality longitudinal quantitative data on individuals and communities.

- I am also not convinced of the analysis results. The Quickbird images used are described as high resolution 60 cm images, and only in the Supplementary Information it is described that the data are 2.4 m multispectral and 61 cm panchromatic, the latter used to pansharpen the former – this means that, while at the time of the disaster these data were indeed novel and considered high-resolution, their level of detail not really sufficient to map specific classes such as foundations, buildings and rubble, let alone what are actually landuse classes (agriculture for example). The limitations of Quickbird images have become clear in other disasters around this time, such as the 2006 Bantul earthquake.

We have removed the initial description that noted a resolution of 60 cm (given that the multispectral images are lower resolution than 60 cm) and apologize for clouding the issue. We agree that accurate classification of rubble and foundation is challenging and we described the difficulties in the revised paper. With respect to buildings, a detailed and innovative evaluation of the Quickbird imagery used in structural damage maps produced for the Bantul earthquake notes that visual interpretation of this imagery by experienced image analysts mapped structural damage with high accuracy (Kerle 2010). It appears that in this case analysts were focused on categories (moderate, heavy, and severe) of structural damage, rather than on the semantic classes that we use, and that they did not distinguish other categories that are quite important for our analysis such as water and agriculture. The Bantul disaster study focused on buildings affected by an earthquake, and the degree of damage they sustained. In contrast, our study focuses on changes in land

cover and land use that resulted from a devastating tsunami as well as changes in land cover and land use during the four years of the reconstruction effort. We do not think the Bantul analysis of building damage provides definitive evidence that that our approach is inadequate given that our study has completely different analytical goals.

To that end, the fact that multiple classes that we create are strongly predictive of a wide range of outcomes at the individual and community levels suggests that our measures contain important signals of damage and recovery, which was our objective. If they contained insufficient information, they would not exhibit statistically significant relationships with the outcomes we consider.

Possibly the imagery available from instruments in use in 2025 could provide even more resolution on post-disaster damage, but these instruments were not in use in 2004 and so cannot provide imagery from the period we analyze: 2004-2009. We analyze data from the 2004-2009 period using the highest quality and highest resolution imagery available for that period: the Quickbird imagery. We are gratified that the reviewer makes this point in their comments.

- Another problem is that in this study a tsunami is equated with a flood (L49, “We focus on flooding”. Sure, both involve water, but they are as different as a storm from a tornado in terms of destructiveness and the resulting damage picture. Tsunamis, and also storm surges (see damage and recovery analysis of post-Haiyan Tacloban), do not simply damage buildings, but create continuous debris fields with floating debris filling most of the space between buildings, often pushing material into and onto building parts. Separating actual buildings, and identifying their damage state, within such a mess is nearly impossible, even in highly detailed drone images. Quickbird cannot reliably reveal actual damage, which is why I have little confidence in the analysis results shown in Fig 2 (panels for 2004DEC30 and 2005AUG06) - likely the actual amount of destroyed buildings is lower.

We appreciate the comment regarding flooding and have revised our discussion of the hazard to more clearly delineate the force of the tsunami and that floods resulting from extreme rainfall or mild storm surge will result in different patterns of damage. The analyses of post-Haiyan Tacloban suggest that, in some cases, strong storms do generate patterns similar to those we observe in the post-tsunami imagery.

With respect to building damage, we did not attempt to classify the damage state of individual buildings or to calculate the number or proportion of buildings that were destroyed. Our approach focused on differentiating patterns that appeared to be buildings from patterns that appeared to be rubble, foundation, water, road, or several other classes (beach, cloud, agriculture), including “other.” In the 2004DEC30 image, relative to 2004JUN23, much less land cover is classified as buildings or roads, and much more is classified as water, rubble, foundations, and “other” (denoting that we were unable to reliably classify it, possibly because of increases in debris). By 2005AUG06 some roads have reappeared, while water and rubble have decreased.

To evaluate the significance of these changes, across many communities, we examine whether the community-level changes in land cover predict mortality, displacement, levels of post-traumatic stress, perceptions of economic resources, and population size. The

answer to that question is an unambiguous affirmative. This would not be the case if our land cover analysis was dominated by noise (from measurement error, say) rather than signal. Furthermore, as measurement error increases, the estimates of standard errors of the coefficients on land cover change increase which undermines significance of those effects. That does not characterize this study: the estimated effects are statistically significant.

- A final point of doubt concerns the damage labelling by volunteers. As someone who was closely involved in post-2010 Haiti GEOCAN effort and other crowdsourced damage labelling efforts I do not believe that it can be done reliably, especially in what are still coarse satellite images (for Haiti 1.24m resolution Geosy 2 images were used for relatively simple seismic damage, and it still did not work). But when using those results in the CNN those errors just propagate.

We have clarified in the revision that we did not use volunteers or any form of crowd sourcing to label the images. Each labeler on our US and Indonesian team was rigorously trained. Three labelers were specifically recruited in Aceh, Indonesia, for the task. After an evaluation period, they were hired and paid to work full time on the project. In particular, they completed all of the labeling used for the testing subset of imagery against which the model was evaluated and segmentation performance was measured. Each of the Indonesian labelers reside in Aceh and have extensive local knowledge of the study area, including the destruction from the tsunami and subsequent reconstruction. Each labeler and their work product was thoroughly evaluated to ensure their work was careful and accurate. All labels were carefully reviewed by the U.S.-based study team. This attention to quality control ensured consistency across labelers and produced high quality labelled data.

Reviewer #4 (Remarks to the Author):

The authors choose the 2004 Indian Ocean tsunami as a test case for flooding damage and hazards. I feel that this is not a suitable choice.

First, this tsunami was induced by an earthquake, which is not related to climate change and sea level rise as indicated in the introduction section.

This is a fair point and we apologize for our lack of clarity. We did not intend to suggest that the earthquake and tsunami were related to climate change. Our goal was to suggest that because climate change is increasing exposures to extreme events (including flooding), it is increasingly important to conduct research on how such exposures affect well-being. We study a source of inundation that is not connected to climate change, but we believe that the results shed light on how inundation affects the well-being of populations and individuals. We have revised the text to clarify this point.

Second, the type and nature of damage after tsunami would be very different compared to the more typical flooding events due to weather such as extreme rainfall and storm surges. The

damage due to tsunami is much more severe and extensive and thus is easily mapped. In contrast, the damage is more subtle in typical flooding events, that would require a more refined algorithm for detection.

We agree that in cases of flooding from extreme rainfall and events where impacts are more subtle the imagery is likely to look different from damage resulting from a tsunami. The similarity between tsunami-induced flooding and flooding from storm surges may depend on the intensity of the event. For example the damage in Tacloban, Philippines associated with Typhoon Haiyan was intense has been described using similar terminology as we use (debris, rubble, inundated land) (Sheykmousa et al. 2019).

In light of these points, we have revised the text to describe more precisely the types of hazards and degrees of destruction for which we believe our results are relevant. We note in the last paragraph of the revised discussion section that different types of disasters will create different patterns of damage.

With respect to recovery, which we also consider, the process of reconstructing “built” infrastructure and restarting agricultural enterprises may look similar across multiple types of hazards.

The CNN model was trained for tsunami damage. It is not obvious whether it would work for typical flooding events. The methodology and results are sound for this tsunami event. The method is good for a study of damage and recovery after a disaster such as the destructive tsunami in this case.

Thank you for this comment, we agree that the method is likely to be most appropriate for disasters that are particularly destructive.

However I would not recommend it for typical flooding events. I would like to see use cases in more typical flooding events that are becoming the norms nowadays

We fully agree. We would also like to see the method applied for other types of events, including flooding that results from storm surge and from intense rainfall.

February 12, 2026

C. Kendra Gotangco Gonzales, PhD
Editorial Board Member
Communications Earth & Environment
orcid.org/0000-0002-3436-9813

Martina Grecequet, PhD
Senior Editor
Communications Earth & Environment
Consulting Editor

Dear Dr. Gotangco Gonzales and Dr. Grecequet,

As noted in your email, Reviewer 2 suggested we could include more examples of how the approach we developed can be applied to improve responses to disasters in the future. In response we have added several examples and appreciate the suggestion.

Our discussion section now includes the following text, in which we have developed several additional examples:

These insights have important implications for science and policy. For example, in the aftermath of a disaster, information is at a premium and these methods can help identify communities in greatest need, including those entirely cut off, as well as routes for evacuation and for provision of assistance. Estimation of the size of the affected population can be improved with these methods, which in turn, will provide a better foundation to assess immediate needs and estimate the costs of damage areas affected by the disaster. It will be possible to inform real-time evaluations of assistance as it is delivered in immediately after disasters and guide dynamic allocation of assistance, ensuring that investments are as effective as possible in supporting the affected populations. Over the longer-run, with these methods, it is possible to document the actual timeline of rebuilding of the built environment, including roads, bridges, and housing along with the re-emergence of agricultural cultivation and other sources of livelihoods and provide a rigorous evaluation of the assistance programs. More broadly, merging information on changes in land cover and land use with data on population measures of socioeconomic status and well-being provides a far richer and more comprehensive picture of immediate impacts of a major disaster and the longer-term trajectories of well-being than does either source alone.

Thank you for the chance to revise our paper.

Sincerely,

Elizabeth Frankenberg, PhD
Cary C. Boshamer Distinguished Professor of Sociology